# Navigating regulatory landscape: A qualitative exploration of medical devices and in vitro diagnostic medical devices oversight in Zimbabwe through key stakeholder perspectives

**Charles Chiku**[1]*, **Talkmore Maruta**[2], **Fredrick Mbiba**[3], **Justen Manasa**[4]

1 Regulation and Prequalification Department, World Health Organization, Geneva, Switzerland,
2 Programs, African Society for Laboratory Medicine, Lusaka, Zambia, 3 The Health Research Unit, Biomedical Research and Training Institute, Harare, Zimbabwe, 4 Department of Oncology, Faculty of Medicine and Health Sciences, University of Zimbabwe, Harare, Zimbabwe

* charleschiku@gmail.com

**Data Availability Statement:** Qualitative Data Repository, https://data.qdr.syr.edu/dataset.xhtml?

## Abstract

Medical devices and In Vitro Diagnostics (IVDs) are vital for public health and accessible healthcare. Still, there is an imbalance in high-quality products in Low and Middle-Income Countries (LMICs). Zimbabwe's regulatory framework for medical devices and IVDs is unclear, leading to ineffective compliance and surveillance. As a result, there are knowledge gaps regarding pre-market and post-market regulatory elements to ensure the safety, quality and performance of medical devices and IVDs used in Zimbabwe. Our study aimed to explore the current status of medical devices and IVD regulations in Zimbabwe. Semi-structured interviews were conducted with 12 regulators from the Medicines Control Authority of Zimbabwe (MCAZ) National Microbiology Reference Laboratory (NMRL), Medical Laboratory and Clinical Scientists Council (MLCScCZ) to understand the current status of medical devices and IVD regulations in Zimbabwe. Three participants completed a questionnaire to understand the regulatory landscape in Zimbabwe. Three key informant interviews were conducted with three regulators from the South African Health Products Regulatory Authority (SAHPRA), Tanzanian Medicines and Medical Devices Authority (TMDA), and World Health Organization Regulatory Systems Strengthening (WHO RSS) to learn best practices to create a roadmap for Zimbabwe. We analyzed qualitative data using a thematic analysis. The findings reveal significant deficiencies and gaps in the legal framework for regulating medical devices and IVDs, highlighting the need for a legal framework and the absence of more comprehensive regulations. Regulatory entities face capacity limitations, especially in regulating medical devices and IVDs. Conformity assessment processes, medical devices, IVD classification criteria, and post-market surveillance also represent challenges, highlighting the need for a well-defined framework and regulatory procedures. The Zimbabwean regulatory system pathway is reactive, prompting several regulatory initiatives to address needs. Despite facing challenges, there is recognition of the importance of collaboration among regulatory authorities, emphasizing a shared commitment to improving and

persistentId=doi:10.5064/F6MLUUS6&version=DRAFT.

**Funding:** The author(s) received no specific funding for this work.

**Competing interests:** The authors have declared that no competing interests exist.

strengthening medical devices and IVD regulations for improved patient safety. By advocating for a proactive, comprehensive, and legally sound approach, indicating the potential for collaboration and synergy, this study provides a foundation for well-informed policy recommendations to guide enhancements and build a framework for a resilient, efficient, and transparent regulatory environment in the Zimbabwe and African regions as a whole.

## Introduction

Medical device and IVD regulations oversee the lifecycle of medical devices from the design, manufacturing, and post-market surveillance activities up to decommissioning based on design. They are based on the intended use, function, and risk classification [1–3]. Medical devices are any instrument, apparatus, appliance, software, implant, reagent, or other similar or related article intended to diagnose, prevent, monitor, treat, or alleviate disease. Medical devices also include products that affect the structure or any physiological process within the human body, provided they do not achieve their principal intended action by pharmacological, immunological, or metabolic means. IVD refers to a medical device, whether utilized independently or in conjunction, designed by the manufacturer to examine specimens derived from the human body in vitro, primarily aimed at supplying information for diagnostic, monitoring, or compatibility purposes [4].

The essential principles for the safety and performance of medical devices and IVDs are guidelines and criteria that manufacturers should adhere to when designing, manufacturing, and placing IVDs on the market. These principles are intended to ensure that IVDs are safe, effective, and reliable for use in various healthcare settings. The following are some of the key essential principles for IVDs: Analytical Performance, Clinical Performance, Risk Management: Stability and Calibration, User Interface and Ergonomics, Biological Safety, Performance Evaluation and Validation, Quality Management System, Labeling and Instructions for Use, and Traceability and Unique Device Identification [5].

Medical devices and IVDs are classified regulatory pathways that are set on a risk-based approach from Class A (lowest risk) to Class D (highest risk) [6, 7]. Design, safety, performance, and quality evaluations are risk-class relative. Lower-risk devices face less strict assessments, while higher-risk ones undergo rigorous evaluations. Medical devices and IVD guidance for regulations set at the international level by the International Medical Devices Regulators Forum (IMDRF) were designed to ensure that the manufacturer who places the product on the market is responsible for all activities, safety and performance by mandating manufacturers to follow risk control based on the device's risk category [8]. While regulations must not hinder economic operators, devices must meet the Essential Principles defined by the regulatory authority and detailed in the Global Harmonization Task and IMDRF guidance documents. These organizations advocate for global regulatory convergence [9].

WHO conducted a desk review in 2016 to assess the availability of medical devices legislation globally. Countries were classified based on pre-market, market, and post-market regulatory elements for medical device legislation. In Africa, 40% of countries had no medical devices regulations, 32% had some, and 28% lacked data. Globally, 58% of WHO member states regulate these devices. Zimbabwe was reported as regulating medical device market placement, but data was unclear [10]. This regulatory disparity could lead to inferior devices and limit access to quality-assured ones. Additionally, the assessment did not specify if the IVDs were included in the assessment since medical devices and IVDs have different frameworks.

Zimbabwe, a low-income country in Southern Africa, has inconsistent medical device and IVD regulations [11]. MCAZ regulates medicines and certain items, while MLCScCZ oversees laboratory personnel and registers IVDs for priority diseases. Hubner et al. found that only condoms and gloves were regulated as medical devices and that Zimbabwe did not have a formal medical devices regulatory system [11].

Dacombe et al. studied HIV-Self Testing IVDs regulation in Malawi, Zambia, and Zimbabwe. Both MCAZ and MLCScCZ claimed the mandate to regulate HIV-Self Testing IVDs [12]. The evolution of the regulatory landscape is unclear based on these two studies. To protect public health, regulatory bodies must enforce public health policies that limit access to the market only to safe, effective, high-quality medical devices and IVDs. Inadequate regulations can hinder manufacturers from meeting approval requirements where they are not well defined, restricting access to needed devices.

## Methods

From June to November 2022, using questionnaires and interviews, stakeholders in Zimbabwe were surveyed to comprehend medical device and IVD regulations and understand the current context, structures, experiences, challenges, and opportunities. Using questions adapted from Rugera et al.'s study, the survey appraised medical devices and IVD safety and effectiveness assessments and controls performed before and after marketing [13]. The tools focused on regulatory, pre-market, marketing, and post-marketing aspects.

A yes/no questionnaire was completed by one staff member from each institution (MCAZ, MLCScCZ, and NMRL), covering topics on regulatory authority, legislation, safety principles, conformity assessment, market authorization, and post-market surveillance for medical devices in general, IVD medical devices for human use and veterinary use. The questionnaire was self-administered electronically and sent to the researcher by email.

Semi-structured interviews were conducted with key informants from MCAZ, MLCScCZ, and NMRL. The discussions explored the WHO Global Model Regulatory Framework for Medical Devices, including IVDs [14] and Global Benchmarking Tool (GBT) + Medical devices and IVDs, a recommended medical device regulation system, focusing on the presence and implementation of legal provisions and regulations—interview questions based on these definitions.

- The National Regulatory System defines quality and safety specifications and standards for the products meeting the definition of medical devices and IVDs and associated assessment procedures (covering pre-market and post-market) to guarantee the quality, safety, and performance of medical devices and IVDs placed on the market.

- Market authorization is a legal framework that defines assessment procedures by national authorities or authorized third parties for licensing or registration of medical devices and IVDs that meet conformity assessment specifications and standards.

- Vigilance is the process of systematic monitoring and collection of information and adverse events, wrong performance or harmful incidents associated with the presence or the use of medical devices and IVDs on the market, as well as related issues. This aims to prompt proper action to remove any deficient product or product with a lower benefit-risk ratio and ensure the presence only of quality products throughout their lifespan.

- Market Surveillance and Control procedures are set by regulatory authorities to ensure that the products on the market meet quality, safety, and performance standards with the highest

risk-benefit ratio. It also includes the mandates of the manufacturer to meet post-market surveillance activities.

- A licensing Establishment procedures set by national regulatory authorities define the specifications and quality standards an establishment has to meet to ensure the quality, safety, transparency and consistency of its activities related to medical devices and IVDs used in or exported from the country as well as the procedure for the assessment and licensing of those establishments.

- Regulatory Inspection audits of establishments are a defined procedure by which an operator's activities are assessed according to applicable rules, laws, standards, norms, or requirements and guarantee compliance at a certain time.

- Laboratory testing procedure is the process by which a regulator defines criteria for assessing specific product performances and their compliance with their performance claims.

- Clinical Trials Oversight is the National Regulatory Authority's(NRA) framework to define requirements, assess, authorize, supervise, and end clinical trials.

The GBT indicators were accepted as the benchmark, so the questions were not piloted [15–17]. The study's exploratory nature and semi-structured interviews prioritized flexibility over rigorous piloting of questions.

Furthermore, to understand best practices, frameworks, and recommendations for establishing and improving medical device regulations from other NRAs in the SADC region, semi-structured interviews were conducted with representatives from the South African Health Products Regulatory Authority (SAHPRA) and one from the Tanzania Medicines and Medical Devices Authority (TMDA). Additionally, a WHO's Regulatory Systems Strengthening regulator was interviewed to gain perspective on using the Global Model Regulatory Framework for Medical Devices (GMRFMD) and how member states have been using it to establish and strengthen their medical device regulatory systems.

The researcher (CC) identified himself to participants during interviews, disclosing his identity, qualifications, occupations, gender, background, and training. The researcher did not have a relationship with the participants. As a Regulatory Affairs Professional for Medical Devices and IVDs with five years of experience in the field, the interviewer conducted remote interviews using Zoom or Teams. The researcher explained the purpose of the study to the participants before obtaining informed consent to take part in the study and to use an audio recorder. All the in-depth interviews were conducted in English, audio recorded, and transcribed verbatim.

## Setting and participants

MCAZ, an NRA, regulates medicines, condoms, and gloves as medical devices. MLCScCZ oversees medical laboratory professionals and operations and evaluates IVDs related to HIV, TB, Malaria, and STIs for national tenders. NMRL, the reference laboratory, performs evaluations for market approval from the MLCScCZ. These three institutions are based in Zimbabwe.

Purposive and snowball sampling were used to select participants, including representatives from medical device regulation institutions. Thirteen participants were chosen from MCAZ, MLCScCZ, and NMRL. Potential participants were identified using institutional input, regulatory documents, and local knowledge. Interviews with MCAZ's Medical Devices and Microbiology unit and legal department provided insights into the IVD evaluation process.

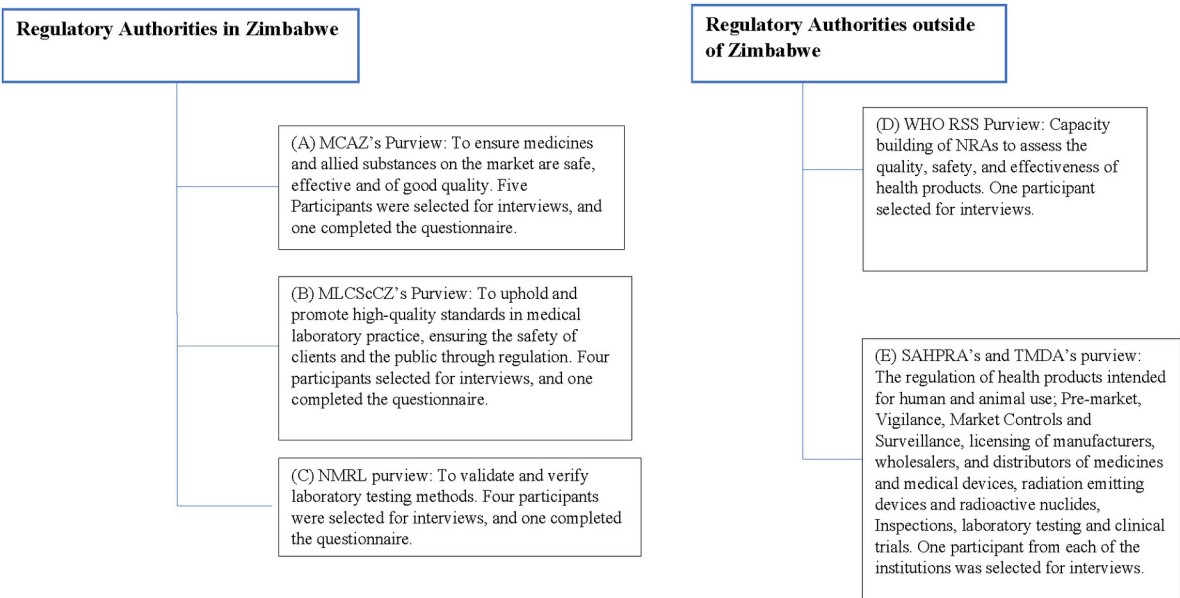

**Fig 1. Flow chart of participants' selection for data collection between June and November 2022.** (A) MCAZ's Medical Devices and Microbiology Unit and Legal Department regulates condoms and Gloves. One participant completed the questionnaire, and five participants were selected for interviews. (B) MLCScCZ registers IVD. Four participants were selected for interviews, including one participant to complete the questionnaire. (C) NMRL conducts performance evaluations of IVDs to aid in the registration of IVDs. Four participants were selected for interviews, including one participant who completed the questionnaire. (D) The WHO's Regulatory Systems Strengthening capacitates national regulatory authorities to strengthen their oversight for safe, quality, and effective medical products. One participant was selected for the interviews. (E)SAHPRA and TMDA are regulatory authorities in South Africa and Tanzania. One participant from each institution was interviewed.

One participant from the SAHPRA and one from the TMDA were interviewed to learn the best practices and lessons in South and East Africa for medical device regulation. The two countries were selected due to their progress in advancing their regulations [11, 18, 19]. A WHO representative discussed the WHO Global Model Regulatory Framework for Medical Devices in Africa and its application in medical device regulations. The plan was to harmonize regional and international medical device regulations in developing Zimbabwe's regulatory system. The sampling flowchart, which outlines the roles of the institutions, is depicted in **Fig 1**.

## Data analysis

The consolidated criteria for reporting qualitative research were used when preparing this manuscript [20]. Data from the questionnaires administered to stakeholders in Zimbabwe was descriptively analyzed to show the availability of regulatory elements at MCAZ, MLCScCZ, and NMRL.

Data analysis from key informant interviews used a thematic analysis framework. The process consisted of deductive analysis supplemented by inductive codes in an iterative process. Key steps in this process included familiarization with the data, achieved through transcribing information and immersing in its content. NVivo 12 software (QSR International) was used for data coding, encompassing open, axial, and selective coding. Theme development involved grouping codes into primary themes and breaking them into sub-themes.

The constant comparison method was applied, entailing ongoing scrutiny of the data to confirm emerging patterns. Thoughts and reflections were documented throughout this

process to develop concepts and theories. Member checking, a validation step, involved sharing findings with participants to align interpretations with their experiences. Synthesis aimed at integrating individual themes and concepts to comprehensively understand medical device regulation in Zimbabwe and the broader region, constructing a coherent narrative of the findings.

The final step, reporting, focused on clearly articulating the findings. The author used quotes and examples to illustrate key points while reflecting on biases and their potential influence on the analysis. CC and FM were involved in the synthesis and analysis process of the data.

### Ethical considerations

Ethics approval was obtained from the Medical Research Council of Zimbabwe, approval number (MRCZ/A/2900). All the participants gave written informed consent to take part in the interviews. Names used in this study are pseudonyms.

## Results

We present the study's findings that aimed to examine the medical device and IVD regulatory landscape in Zimbabwe and lessons learned from SAHPRA, TMDA, and WHO RSS in establishing and strengthening medical device regulations. The results are organized into three sub-sections. The first sub-section presents results from the questionnaires, the second from the semi-structured interviews of stakeholders in Zimbabwe, and the third from interviewing regulators from SAHPRA, TMDA, and WHO RSS. **Table 1** shows the characteristics of the institutions and participants that were interviewed.

### Pre-market and post-market regulatory measures elicited through questionnaire surveys in Zimbabwe

Qualitative data revealed that MCAZ, MLCScCZ, and NMRL had no legal power to regulate medical devices and IVDs as shown in **Table 2**. MCAZ regulated condoms and gloves and drafted regulations for medical devices, including IVDs. The MLCScCZ registered IVDs for HIV, Malaria, SARS-CoV-2 and other priority diseases based on a performance evaluation conducted at the NMRL. IVDs for human and veterinary use, the public and private sectors,

**Table 1. Characteristics of interview participants interviewed in Zimbabwe between June to November 2022.**

| Participant | Institution | Structured questionnaire administered (Y/N) | Interview (Y/N) |
|---|---|---|---|
| KI01 | MCAZ | Y | Y |
| KI02 | NMRL | Y | Y |
| KI03 | MLCScCZ | N | Y |
| KI04 | MLCScCZ | Y | Y |
| KI05 | NMRL | N | Y |
| KI06 | MCAZ | N | Y |
| KI07 | MLCScCZ | N | Y |
| KI08 | NMRL | N | Y |
| KI09 | NMRL | N | Y |
| KI10 | MCAZ | N | Y |
| KI11 | MCAZ | N | Y |
| KI12 | MLCScCZ | N | Y |

*Note*. MCAZ = Medicines Control Authority of Zimbabwe; MLCScCZ = Medical Laboratory and Clinical Scientists Council of Zimbabwe; NMRL = National Microbiology Reference Laboratory.

**Table 2. Summary of medical devices and IVD regulatory elements assessed using the questionnaire.**

| Institution | Legal definition for medical devices and/or IVDs availability. | Utilizing Essential Safety and Performance criteria for market approvals. | Published list of relevant standards in law. | Technical specifications available for specific Medical devices and/or IVDs. | Availability of regulations classifying Medical devices and IVDs. | Availability of published conformity assessments for medical devices and IVDs. | Availability of post-market surveillance policies and requirements. |
|---|---|---|---|---|---|---|---|
| **MCAZ** | No for Medical devices | No | No | No | No for Medical devices | No | No |
| | Yes, for IVDs in the draft IVD regulations. | | | | Yes, for IVDs in the draft IVD regulations. | | |
| **MLCScCZ** | No | No | No | No | No | No | No |
| **NMRL** | No | No | No | No | No | No | No |
| **WHO guidance** | Yes | Yes | Yes | Yes | Yes | Yes | Yes |
| **IDMRF guidance** | Yes | Yes | Yes | Yes | Yes | Yes | Yes |

MCAZ: Medicines Control Authority of Zimbabwe, MLCScCZ: Medical Laboratory and Clinical Scientists Council of Zimbabwe, NMRL: National Microbiology Reference Laboratory, WHO: World Health Organization, IMDRF: International Medical Devices Regulators Forum.

and blood products are unregulated. Draft import and export and medical devices regulations exist, but no conformity assessment, marketing controls, or post-market monitoring exist. The MCAZ has included a definition of Medical devices and IVD medical devices in the draft import and export regulations and IVD regulations. MLCsCZ and NMRL do not have documented definitions for medical devices, including IVDs. Furthermore, no rules exist to classify medical devices, including IVDs, into risk classes at the three institutions. There are no documented principles to ensure the safety and performance of medical devices, including IVDs. The MCAZ has a policy on reliance and recognition mechanisms to leverage approvals from the WHO Prequalification of In Vitro Diagnostics. "*We have a policy on Reliance, specifically relying on the WHO prequalification programme*" (MCAZ KI01). In contrast, the MLCScCZ and NMRL do not have reliance and recognition mechanisms. MCAZ had included mechanisms for emergency authorization or listing in case of public health emergencies in the draft regulations. In contrast, the MLCScCZ and the NMRL did not have this mechanism. There are no mechanisms to declare conformity to specified requirements nor to prohibit deceptive, misleading, or falsified advertising for medical devices, including IVDs. Lastly, there is no documentation and implementation of a post-market surveillance system.

## Findings derived from stakeholder interviews in Zimbabwe: A comprehensive examination of the present landscape, frameworks, experiences, challenges, and prospects

Twelve participants were interviewed, yielding a 92.3% response rate (one refusal). Data were categorized into regulatory system components, revealing six main themes after transcript analysis, as shown in **Table 3**: legal framework, regulatory capacity, conformity assessment, post-market surveillance, medical device regulation, and collaboration of regulatory authority.

### Legal framework

Zimbabwe's medical device regulations are inadequate. MCAZ uses the Medicines and Allied Substances Control Act (MASCA) to regulate condoms and gloves, while the MLCScCZ uses

**Table 3. Primary themes arising from interviews with regulatory stakeholders in the medical device sector in Zimbabwe.**

| Theme | Summary Findings |
|---|---|
| Legal framework | The MCAZ lacked a legal basis to regulate medical devices. The MLCScCZ attempted to use the Health Professions Act to regulate IVDs, but it was not applicable. |
| Regulatory capacity | The MCAZ and MLCScCZ lacked technical expertise, funding, and infrastructure to regulate medical devices, including IVDs. The NMRL lacked technical expertise, competencies, and reference materials to verify IVD safety and performance, which the MLCScCZ was responsible for. |
| Conformity assessment | Glove and condom conformity assessment was identical despite different risk classes. |
| Post-market surveillance | No post-market surveillance is needed for public safety. |
| Medical devices regulatory system | A balanced, risk-based IVD regulatory system must be established and strengthened. Training and technical support, infrastructure, trained personnel, and transparency and accountability mechanisms are needed. |
| Collaboration of regulatory authorities | MCAZ, MLCScCZ, NMRL, and other stakeholders must collaborate using systems thinking to ensure IVD regulations are risk-based, responsive, and effective. |

*Note.* MCAZ = Medicines Control Authority of Zimbabwe; MLCScCZ = Medical Laboratory and Clinical Scientists Council of Zimbabwe; NMRL = National Microbiology Reference Laboratory.

the Health Professions Act to regulate IVDs for priority diseases. MASCA does not cover all regulatory aspects, such as product scope and the mandate to regulate medical devices. Zimbabwe has no law requiring safe, high-quality medical devices. MCAZ regulates condoms and gloves as medical devices, noting, *"No laws exist for medical devices, but condoms and gloves are regulated using the medicines framework"* (MCAZ KI07). The MLCScCZ was registering IVDs for priority diseases for the national tendering process. "W*e had no power but approved products based on the experts' committee data and performance evaluations"* (MLCScCZ KI04). The Health Professions Act, employed by MLCScCZ, lacks provisions pertaining to the oversight of IVDs. Although MCAZ has drafted import/export regulations for medical devices, awaiting approval, it has also expressed intentions to amend the MASCA to encompass the regulation of all health products. MCAZ contemplates a name change to reflect a broader scope beyond medicines, a matter that has been discussed within the Ministry of Health. The regulatory framework for IVDs under the NMRL remains ambiguous, with an interviewee acknowledging the difficulty in locating the specific Act of Parliament delineating NMRL's IVD regulatory functions (NMRL KI08).

## Regulatory capacity

At the time of the interviews, the MCAZ only regulated condoms and gloves. It has a limited capacity to regulate IVD medical devices. They were engaging the MLCScCZ to form an expert committee to help with IVD regulatory functions in the future: "*The technical committee approves, then external stakeholders are consulted, including MLCScCZ, Health Professions Authority, and the Ministry"* (MCAZ KI01). The MLCScCZ acknowledged its difficulty regulating IVDs and wanted to collaborate with MCAZ, which has experience with medicines, condoms, and gloves: "*We'll inspect manufacturing sites like MCAZ does for drugs"* (MLCScCZ KI03).

The NMRL faced challenges such as untrained staff on regulatory affairs, the absence of Clinical and Laboratory Standards Institute (CLSI) guidelines and specimen testing panels, and non-standardized protocols. They relied on the results of other institutions, such as the

National Blood Services Zimbabwe. Regulatory bodies lack the resources and knowledge to regulate medical devices such as IVDs effectively.

### Conformity assessment

The MCAZ assesses condoms and gloves, but there are no criteria for classifying medical devices to determine the level of scrutiny needed, *"At the moment we do not have medical device classification rules published. However, we have included them in the Import and Export of Medical Devices Regulations 2017"* (MCAZ KI07). Some gloves are Class A and are regulated with the same stringency as Class C devices: *"Gloves' conformity assessment is not tied to their risk Class A. We plan to evaluate medical devices based on the risk class in the draft regulations. High-risk products will undergo a stricter conformity assessment for market access."* (MCAZ KI11).

### Post-market surveillance

The MCAZ only conducted routine inspections for gloves and condoms, not formal post-market surveillance. They admitted that this was a weakness. MLCScCZ receives IVD reports, but the process is undocumented. There is no documented post-market surveillance obligation for economic operators. MLCScZ KI03 stated, *"There is no clear guidance on the process for manufacturers, representatives, distributors, or importers from a regulatory and economic standpoint."*

### Medical devices and IVDs regulatory system

It was found that the medical devices and IVDs regulatory system in Zimbabwe was not proactive, effective, or proportional to the risk class of medical devices and IVDs. Participants concurred that medical devices and IVDs regulation needs enhancement. *"Existing regulation is inadequate, primarily responding to national disease programs like HIV/AIDS, TB, and Malaria, leaving other IVDs unregulated" MLCScCZ KI12).* Additionally, condoms were being regulated due to the HIV/AIDS pandemic. *"We test for gloves though there is a low risk because of HIV, and condoms are tested because we have been hit so badly"* (MCAZ KI1). The MCAZ identified the gap in the current regulation of medical devices, resulting in the drafting of Import and Export of Medical Devices regulations, followed by IVD Medical Devices Regulations. MCAZ KI07 commented, *"MCAZ is drafting regulations for medical device import and export control to comprehend the market size and regulate high-risk devices as per IMDRF guidance."*

### Collaboration of regulatory authorities

The MCAZ and MLCScCZ collaborate, recognizing each other's roles in medical device regulation. The MCAZ acknowledges the NMRL's role in regulatory laboratory testing, allowing them to continue testing on the regulator's behalf. A respondent commented, *"There is a need for collaboration, infrastructure, and leadership, with patient safety as the priority"* (MLCScCZ KI11). These responses demonstrate the stakeholders' willingness to collaborate and improve medical device regulations.

### Insights gained from regulatory authorities: Perspectives derived from interviews with SAHPRA, TMDA, and WHO RSS representatives

WHO, SAHPRA and TMDA regulators were interviewed to learn best practices that can be used to make recommendations on the medical devices roadmap for Zimbabwe. Insights were gained from these interviews, and interviewees' characteristics and themes are summarized in **Tables 4 and 5**.

**Table 4. Characteristics of key informants that participated in interviews outside Zimbabwe.**

| Key Informant | Institution |
|---|---|
| WHO KI13 | WHO |
| SAHPRA KI14 | SAHPRA |
| TMDA KI15 | TMDA |

**Note** SAHPRA = South African Health Products Regulatory Authority; TMDA = Tanzania Medicines and Medical Devices Authority

## International standards and tools

The WHO GMRFMD is a reference for medical device regulations. "*In a survey of African countries, 60% were aware of the framework and used it as a guide for regulating medical devices and IVDs*" (WHO KI13). WHO GMRFMD is a viable framework for countries to create and strengthen medical device regulations. South Africa has employed the WHO Global Benchmarking Tool (GBT) to improve its medical device regulatory system. The GBT has indicators to measure the system and to create a trackable plan. There are plans to revise the WHO GMRFMD. The WHO representative said, "*The GBT indicators prompted discussions to update the model to include GBT + medical indicators*"(WHO KI13).

## Streamline legislation

South Africa and Tanzania have had to modify their legal systems to regulate medical devices. "*The Medicines Control Council (MCC) lacked the authority to do so before 2017, but industry pushback led to an update of the legislative framework*" (SAHPRA KI14). The Tanzanian NRA representative commented on the legal framework change, "*The minister requested minor modifications to the law to include IVDs. This amendment included device risk classes and granted ministers power to make other regulations. This clause was used to create regulations for medical devices signed by the minister in 2015*" (TMDA KI15). The legislation establishes regulations and enforces regulations.

**Table 5. Main themes emerging from interviews with key informants outside Zimbabwe.**

| Theme | Summary of the finding |
|---|---|
| International standards and tools | The GMRFMD is a reference document states use to strengthen medical device regulations. It is being updated to include GBT plus indicators and advanced regulatory elements. SAHPRA used the GBT to evaluate its regulatory capacity and develop roadmaps to address identified deficiencies. |
| Streamline legislation | SAHPRA and TMDA amended laws to allow NRAs to regulate IVDs, making regulations enforceable. |
| International and regional harmonization | Harmonization is essential to reduce regulation and share knowledge. |
| Stakeholder engagement and advocacy | Strategic stakeholder engagement is essential, considering each group's needs and dynamics. Open communication and stakeholder involvement in decision-making can improve regulatory systems. |
| The transition period to implement new regulations | Manufacturers and other economic operators need a transition period to comply with new medical devices, including IVDs, regulations to avoid access disruptions. |

**Note**: WHO GMRFMD = World Health Organisation Global Model Regulatory Framework for Medical Devices; GBT = Global Benchmarking Tool, SAHPRA = South African Health Products Regulatory Authority; TMDA = Tanzania Medicines and Medical Devices Authority

### International and regional harmonization

Tanzania aligned its regulations with the regulations from the United States of America Food and Drug Administration (USFDA), the Australian Therapeutic Goods Administration (TGA)and the Japanese Pharmaceutical and Medical Devices Agency(PMDA). The aim was to harmonize Tanzania's medical device regulations. They referenced these mature jurisdictions for their advancement in this field. *"The Tanzanian regulatory authority shaped its regulations following the IMDRF framework to ensure harmonization, mirroring the medical regulatory framework"* (WHO KI15).

The SAHPRA representative suggested that Zimbabwe joined harmonization initiatives, "*such as the African Medical Device Forum and Global Harmonization Working Party, to learn from other regulatory authorities, which is a good information-sharing platform. This can help Zimbabwe harmonize "* (SAHPRA KI14).

### Stakeholder engagement and advocacy

Tanzania's medical device regulations were shaped by stakeholder engagement and advocacy from technical departments to policymakers. "*We advocated and informed the Ministry of Health's top officials, such as the minister, so that they can understand and fix any mistakes."* (WHO KI13). The legislation clearly defines mandates in South Africa, simplifying engaging with stakeholders in regulating medical devices. The participant commented, *"The mandate did not permit it." For example, the Act covers radioactive medical devices and includes a national regulatory body. However, the Ministry of Health could not regulate the health products, isotopes, and radiation from Eskom. The legal framework barred them from regulating medical devices and IVDs"* (SAHPRA KI14). Interviews demonstrated that stakeholder engagement and advocacy are necessary to understand medical device regulation and leadership involvement.

### The transition period to implement new regulations

Regulations should not impede access to medical devices and IVDs when they are introduced. For this reason, countries such as South Africa and Tanzania allowed a two-year transition period for economic operators to comply with new regulations, avoiding disruptions in medical device access. *"In 2019, companies had to register or be shut down. SAHPRA sent acknowledgment letters to 600–700 applicants annually"* (SAHPRA KI14). Tanzania allowed IVDs registered by the PHLB to access the market and allowed manufacturers a transition period to register their IVDs. *"Initially, the population had to rely on the previous regulatory approval by the PHLB before full registration was allowed. Although grandfathering was acknowledged, regulations covered all aspects, including site performance"* (TMDA KI15).

## Discussion

### Current regulatory landscape in Zimbabwe

The study revealed significant gaps in the regulatory framework for medical devices and IVDs among the three institutions—MCAZ, MLCScCZ, and NMRL. MCAZ, although responsible for regulating condoms and gloves, lacked legal authority to regulate medical devices and IVDs, although it drafted regulations for them. On the other hand, MLCScCZ registered IVDs for priority diseases based on performance evaluations conducted at NMRL. However, IVDs for both human and veterinary use, as well as blood products, remained unregulated across public and private sectors. While draft regulations for import/export and medical devices exist, there were no conformity assessment, marketing controls, or post-market monitoring procedures.

The importance of medical device regulations upholds safety, quality, and performance standards for public health [8]. Effective regulations necessitate a legal framework outlining the national regulatory authority's mandate, system, range of regulated products, and regulatory functions. The current regulations lack legal definitions of products and entities to regulate [15]. Condoms and gloves met conformity assessment standards for approval. However, other medical devices like IVDs, microscopy stains, and veterinary devices are unregulated. Despite the MLCsCZ registration of HIV, Malaria, TB, Syphilis, and COVID-19 IVDs through performance evaluations, the approach is ineffective. Conformity assessment ensures the safety, quality and performance of IVDs. Due to limited capacity, IVDs' performance was partially verified without technical documentation, manufacturing site inspections, or labelling reviews. Thus, Zimbabwe does not have a formal medical device regulatory system, similar to the finding of Hubner et al. [11].

The MLCScCZ regulated IVD and coordinated IVD registration for public tenders due to the health minister's worries over unregulated HIV test kits. The MCAZ, lacking legal authority, regulated gloves and condoms as IVDs due to the HIV pandemic. It is uncertain if authorities understand the need for an effective, responsive, and proportional regulatory system. Authorities in medical device and IVD regulation play a pivotal role in shaping policies, ensuring effective oversight, fostering collaboration, adapting to technological changes, and prioritizing public health. Their decisions directly impact the safety and efficacy of medical devices and the overall well-being of patients and the healthcare system.

The MLCScCZ acknowledged its inability to regulate IVDs and attributed this responsibility to MCAZ. However, despite possessing regulatory powers, MCAZ did not regulate IVDs, resulting in a regulatory gap. To address this, it was recommended that MCAZ take prompt action to enhance the regulatory framework. A unified regulatory approach was emphasized to streamline enforcement.

This challenge of IVD regulation was evident in a study by Dacombe et al. on HIV self-testing IVD regulation in Malawi, Zambia, and Zimbabwe. The study revealed confusion regarding the responsible regulator in Zimbabwe, with MCAZ and MLCScCZ claiming jurisdiction [12]. To resolve this, MCAZ proposed the establishment of a single regulatory body for IVDs, with collaboration from MLCScCZ. MLCScCZ expressed its intention to collaborate with MCAZ in IVD regulation by forming an advisory committee. Additionally, there is a need for clarity regarding NMRL's authority in conducting market authorization testing for IVDs. Legal measures are deemed necessary to empower NMRL to carry out testing for IVDs officially.

Before selling a medical device and IVDs, a conformity assessment is required according to IMDRF based on the device's risk class. Post-market surveillance confirms safety and performance standards, preserving public trust after the product's approval for marketing. The assessment should match the device's risk classification. The MCAZ assessed glove and condom dossiers to validate manufacturers' claims. It conducted analytical and clinical evaluations, manufacturing site inspections, and product performance tests. The rationale for MCAZ's regulation of low-risk Class A and B gloves is unclear, while Class C condom regulation for contraception or STI prevention is logical. Risk classification should match product risk, lower regulatory burden, and expedite product access [6].

The MLScCZ registers IVDs after NMRL evaluation. NMRL staff lack regulatory training and cannot access CLSI's IVD protocols. NMRL also lacks International Standard reference panels for HIV and TB tests, to mention a few. NMRL only evaluated diagnostic sensitivity and specificity. It failed to assess the Limit of Detection, precision, cross-reactivity with endogenous/exogenous substances, cross-reactivity with other microbial agents, specimen stability, and IVD stability. Thus, NMRL-registered IVDs may be unreliable.

Zimbabwe's law does not permit reliance and recognition mechanisms. Reliance and recognition mechanisms involve an NRA deciding on product approval, even if it relies on other authorities. The relying authority is independent and accountable for decisions. Regulatory reliance can reduce access barriers and use resources efficiently. It can reduce uncertainties for innovators and improve crisis responses. The approach speeds up access to safe, quality health technologies [21].

The WHO prequalification program assesses the safety, quality, and performance of IVDs, including technical documentation reviews, site inspections, performance testing, and labelling reviews [22]. The Collaborative Registration Procedure, a reliance mechanism, fosters information sharing among the WHO prequalification, NRAs, and manufacturers, speeding up the registration of quality-assured IVDs and reducing regulatory redundancy [23]. It also considers LMIC, enhancing the accessibility of safe, quality IVDs. This mechanism could quicken IVD registration in Zimbabwe but requires the MASCA amendments to include reliance and recognition mechanisms.

Post-market surveillance documentation is lacking, with the MCAZ only monitoring condoms and gloves in the market. While the drafted regulations for medical device import and export and IVD include some post-market surveillance elements, they are not comprehensive. This aligns with the findings of Hubner et al. [11]. Both Dube-Mwedzi et al. and Stanislav et al. have highlighted insufficient capacity and effort in post-market surveillance, specifically in the prevention, detection, and response to substandard and falsified medical products [24, 25]. Our study confirms a correlation with the findings of Stanislav and Mwedzi-Dube, though the studies were for the whole of Southern Africa, Zimbabwe included. Consequently, there is a pressing need for substantial progress in establishing an effective post-market surveillance system.

Participants stressed the need for cooperation and improved medical device regulations. A participant noted that disagreements resulted in isolated work and professional conflicts. Laboratory scientists see IVDs as tools of their trade that must be regulated by the laboratory professionals council, leading to a stalemate between MLCScCZ and MCAZ sometime back. Collaboration, which utilizes collective skills and abilities, is crucial for effective problem-solving, innovation, and growth. It improves communication, fosters relationships, and aids in achieving common goals, making it vital for efficient medical device and IVD regulation. It is high time for institutions to start collaborating for public health.

A roadmap to address the abovementioned gaps requires qualified, competent, skilled professionals. A system-thinking approach involving practitioners, researchers, advocates, industry representatives, and leaders is crucial. Strategic stakeholder engagement, considering each group's needs, is necessary. Open communication and stakeholder participation in decision-making can enhance projects and relationships. Understanding system organization, dynamics, networks, and knowledge is vital for effective medical device and IVD regulation. A solution requires cooperation and a systems-based approach to ensure an effective regulatory framework. As shown in **Fig 2**, implementing such an approach will help Zimbabwe develop a successful plan to oversee medical devices.

**Exploring opportunities for improvement.** GMRFMD is essential for NRAs to enhance medical device regulations, particularly in developing countries. The framework should include GBT+ medical device indicators. SAHPRA revised its regulations using GMRFMD and WHO GBT and compared its review processes with other NRAs for improvement and harmonization. This led to the MCC's rebranding to SAHPRA, allowing it to regulate medical devices, including IVDs. The MCC followed Good Review Practices but took longer to grant market authorization when it compared its practices with other established NRAs [18, 19, 27].

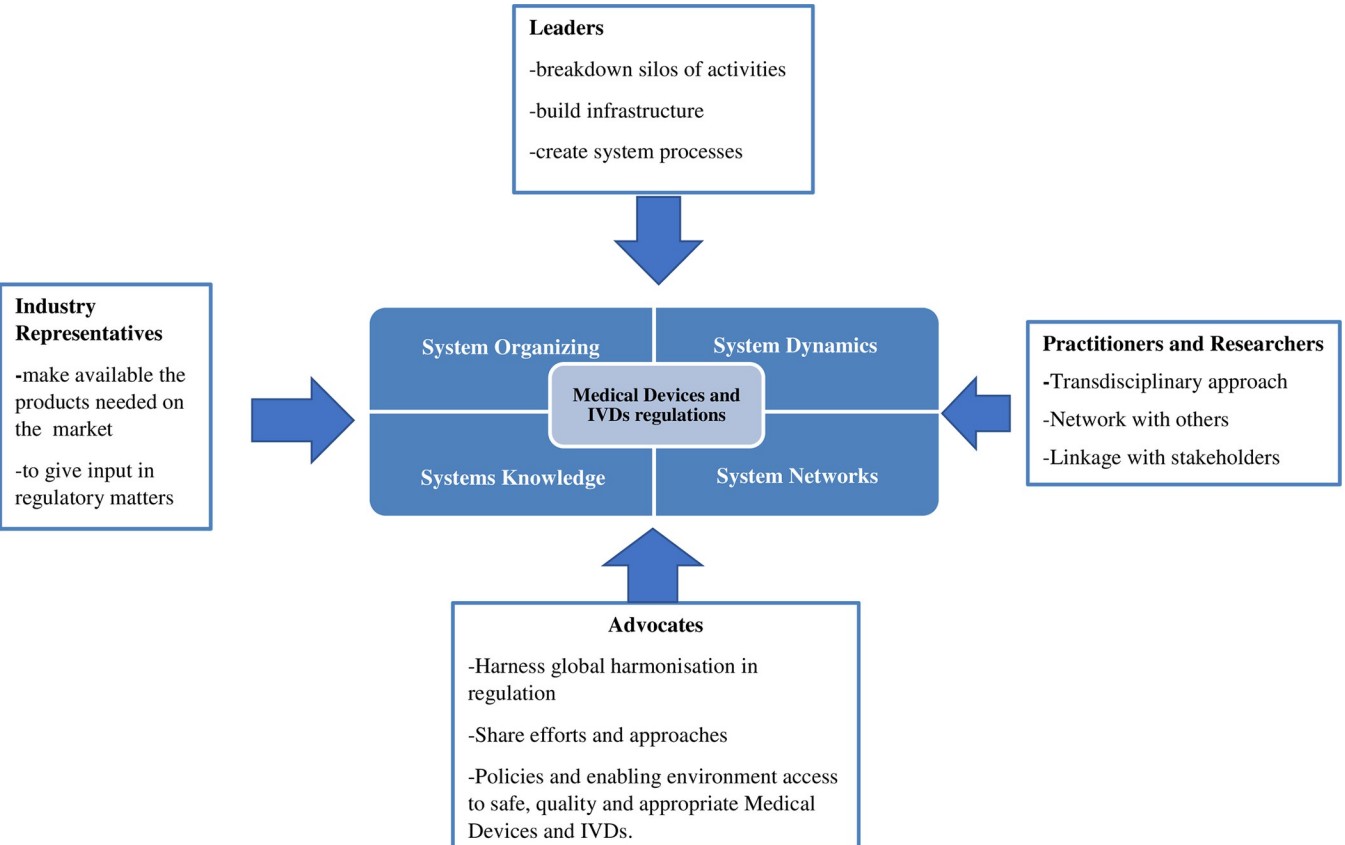

**Fig 2. Proposed stakeholder engagement and systems thinking approach for Zimbabwe medical devices and IVDs regulations.** Researchers, practitioners, leaders, and advocates are critical stakeholders in formulating and implementing medical devices and IVDs regulations. The integrated approach for effective regulation requires: Systems Organization: The organization of systems aims to comprehend and advance collaborative frameworks responsible for the regulation, facilitation, management, and promotion of medical devices and IVDs. This includes fostering systematic actions and facilitating continuous learning within the regulatory domain. System Dynamics: System dynamics play a crucial role in modeling the complex interactions within the regulatory system. This involves capturing the intricacies of factors influencing medical device utilization, such as legislative frameworks, research endeavors, control activities, industry dynamics, and socio-cultural factors. System Networks: System networks analyze collaborative relationships within the regulatory landscape. Their purpose is to enhance strategies for collaboration and minimize redundancy within the systems that oversee medical devices and IVDs. Systems Knowledge: System networks contribute to analyzing collaborative relationships, improving collaboration strategies, and mitigating redundancies within the broader context of medical device and IVDs regulation [26].

Tanzania's NRA has adopted best practices from USFDA, TGA, and PMDA for regulatory alignment and establishing reliance mechanisms [13]. In the manuscript's discussion, it is imperative to acknowledge the crucial role of stakeholder involvement in achieving regulatory harmonization. The participation of policymakers, regulators, and economic operators is indispensable for garnering regulatory acceptance. Harmonization and convergence of medical device regulations, as seen in South Africa and Tanzania, are vital. Convergence adopts international guidelines, while harmonization standardizes technical guidelines, easing the regulatory load on manufacturers and regulators [28]. Keyter et al. [27] suggested efficient regulatory pathways and risk-based reviews to save resources and ensure device availability. **Table 6** shows changes required in Zimbabwe's Medicines and Allied Substances Control Act based on the best practices learned from South Africa and Tanzania.

TMDA lacked medical device and IVD experts, leading to transition regulation issues. In regulating medical devices and IVDs spanning various complexities and functionalities, the

**Table 6. Recommendations on the amendment of the medicines and allied substances control act.**

| Section to be Amended | Details of the amendment |
|---|---|
| Institutions with the mandate to regulate medical devices, including IVDs. | Identify the institutions with legal authority to regulate medical devices and enforce regulations and market oversight. |
| Definitions of regulated health products | This law defines medical and veterinary devices, including IVDs, and outlines the scope of medical devices regulated. It is flexible enough to accommodate new technologies and treatments. |
| Stakeholder responsibilities | Stakeholders have the authority to issue regulations and act when health is at risk and must publish guidance to help understand legal requirements. |
| Reliance and Recognition Mechanisms | Give regulatory authority discretion to rely on and recognize other jurisdictions' work or decisions. |
| Explicit requirements for products to access the market | Regulations require only safe, quality, and effective medical devices and IVDs that meet conformity assessment using a risk-based approach (conformity assessment proportional to the medical device's risk class.) |
| Registration requirements | Establish record-keeping, registration, and reporting requirements for all parties within the scope of the law. |
| Licensing Establishment | Establish record-keeping, registration, and reporting requirements for manufacturers, distributors, importers, wholesalers, and authorized representatives in all the value chains of medical devices, including IVDs, within the scope of the law, including regulatory authority. |
| Post-market Surveillance | Set up requirements with clear roles, communication, and feedback loops to monitor medical devices' safety, quality, and performance until the product's decommissioning. Post-market surveillance must be proportional to the device's risk class, including IVDs. |
| Laboratory Testing | Requirements for laboratory testing for medical devices, including IVDs, are proportional to the risk class of these products. |
| Clinical Trials, | Regulations for investigational IVDs shipped in-country before marketing approval. |
| Donations | Requirements that donated medical devices, including IVDs, must meet before acceptance into the country. |
| Transition Period | Allow enough time for those affected by the law to comply and ensure minimal interruption to medical device supply to health facilities and users. |

collaboration of multidisciplinary professionals is paramount. Their diverse expertise, encompassing engineering, medicine, biology, regulatory affairs, and quality assurance, ensures comprehensive evaluation, robust standards adherence, and effective oversight, safeguarding public health and promoting innovation in healthcare technologies. The TMDA formed an external expert committee to guide medical device aspects, and it has grown to a directorate with 40 staff members doing regulatory and conformity assessment work.

The TMDA had to amend medicine regulations to regulate medical devices, including IVDs. Medicine's approach may not apply to medical devices. For medicines, regulatory standards often stipulate that a certain percentage of the stated shelf life must remain upon arrival at the entry port. This requirement ensures that the product maintains its potency and efficacy for a specified duration after reaching the market. For example, medicines with a longer shelf life, such as 36 months, are expected to have a higher percentage (e.g., 80%) of that shelf life remaining upon importation.

In contrast, the regulatory expectations for IVD reagents and controls differ significantly, especially those with short shelf lives (less than 6 weeks). These products may be permitted to arrive with a much shorter remaining shelf life, such as only two weeks, due to the nature of

their use and stability characteristics [29]. This allowance acknowledges the inherent limitations in the shelf life of certain IVD components, which may degrade more rapidly than pharmaceuticals.

Understanding these distinctions is crucial for regulatory compliance and importation processes, as they reflect the tailored requirements and considerations specific to medicines and IVDs within the regulatory framework.

They suggested that the NRA create a framework based on medical device design and use, including IVDs.

The medical device regulatory system needs a competency framework to identify key competencies, levels, and domains. Levels 1–4 range from operational to strategic tasks, with Level 4 professionals expected to perform Levels 1–3 tasks if necessary. This framework should include scientific and health concepts, ethics, business acumen, communication, leadership, regulatory frameworks, product lifecycle strategy, product development and registration, post-approval changes, and post-market surveillance [30].

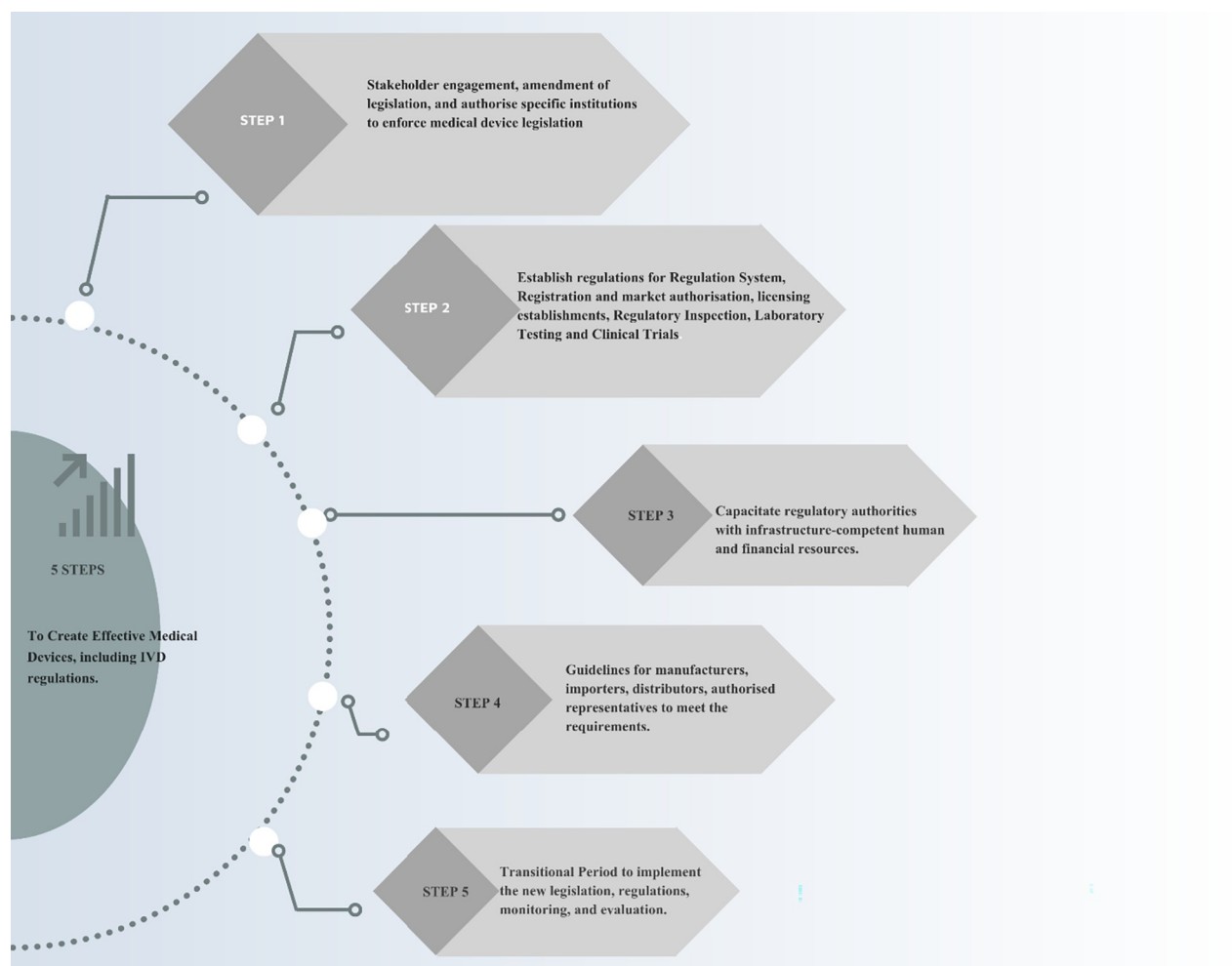

**Fig 3. Proposed roadmap to create effective medical device and IVD regulations.** Step 1: Change of legislation, Step 2: Establishment of regulations for the regulatory functions of a national regulatory authority, Step 3: Resourcing of the regulatory authorities, Step 4: Guidance development to aid economic operators to comply with regulations, Step 5: Transition period for implementation of new regulations, monitoring, and evaluation of implementation of the regulatory system.

Government funding is essential for the regulatory system to work properly. Inadequate funding may lead to rushed product registration. The system needs trained staff, infrastructure, facilities, and information technology. Resources should be allocated according to the law, with the capacity to increase as the system develops [14].

The study proposes a three-tiered approach to establishing and improving a transparent regulatory system for Zimbabwe. This includes legislation, regulations, and guidance. Legislation, under public scrutiny, must align with the constitution. Regulations interpret the law for implementation, and guidance helps manufacturers comply with regulations [3]. The proposed roadmap is illustrated in **Fig 3.**

## Study limitations

This study has several limitations. It excludes views from various stakeholders in Zimbabwe's medical devices and IVD regulations, such as health officials, users, donors, and manufacturers. The research strategy limits the wider applicability of the results, with a focus on reliability over generalizability. Gradual generalization may occur with more empirical research. The study's standardized scope prevented triangulation. The tools used in the studies were not piloted due to the small sample size, which prevented questionnaire testing and methodology piloting. Additionally, our study used a questionnaire and interviews to bypass the pilot phase and ensure implementation confidence. Our study is grounded in well-established regulatory practices, minimizing the risk of not being tested.

## Conclusions

In conclusion, the study underscores the multifaceted challenges within Zimbabwe's medical devices and IVD regulatory frameworks and provides valuable insights for advancing the regulatory landscape. Regulatory bodies' identified gaps and collaborative intentions set the stage for future regulatory enhancements, emphasizing the imperative for a proactive, comprehensive, and legally sound approach to medical devices and IVD regulation in Zimbabwe. The study's findings are a foundation for informed policy recommendations and actions to foster the country's robust and effective medical devices and IVDs regulatory environments. Harmonizing regulations with other regional and international regulatory authorities, reliance, and recognition mechanisms must play a pivotal role in developing the roadmap to strengthen the medical devices regulatory system.

## Acknowledgments

The authors would like to acknowledge the cooperation of the agencies and personnel interviewed in Zimbabwe, South Africa, Tanzania, and the WHO for their assistance in accessing documentary evidence and reports.

## Author Contributions

**Conceptualization:** Charles Chiku.

**Data curation:** Charles Chiku, Fredrick Mbiba.

**Formal analysis:** Charles Chiku.

**Investigation:** Charles Chiku.

**Methodology:** Charles Chiku.

**Project administration:** Charles Chiku.

**Resources:** Charles Chiku.

**Supervision:** Talkmore Maruta, Justen Manasa.

**Validation:** Fredrick Mbiba.

**Visualization:** Charles Chiku, Talkmore Maruta, Fredrick Mbiba, Justen Manasa.

**Writing – original draft:** Charles Chiku, Talkmore Maruta, Fredrick Mbiba, Justen Manasa.

**Writing – review & editing:** Charles Chiku, Talkmore Maruta, Fredrick Mbiba, Justen Manasa.

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
