## [Decision Letter · Decision Letter 0]

18 Aug 2023

PONE-D-23-17224Regulation of Medical Devices in Zimbabwe: A qualitative study with key stakeholdersPLOS ONE

Dear Dr. Chiku,

Thank you for submitting your manuscript to PLOS ONE. After careful consideration, we feel that it has merit but does not fully meet PLOS ONE’s publication criteria as it currently stands. Therefore, we invite you to submit a revised version of the manuscript that addresses the points raised during the review process.br /> Please submit your revised manuscript by Oct 02 2023 11:59PM. If you will need more time than this to complete your revisions, please reply to this message or contact the journal office at plosone@plos.org. Please include the following items when submitting your revised manuscript:A rebuttal letter that responds to each point raised by the academic editor and reviewer(s). You should upload this letter as a separate file labeled 'Response to Reviewers'.A marked-up copy of your manuscript that highlights changes made to the original version. You should upload this as a separate file labeled 'Revised Manuscript with Track Changes'.An unmarked version of your revised paper without tracked changes. You should upload this as a separate file labeled 'Manuscript'.

We look forward to receiving your revised manuscript.

Kind regards,

Adetayo Olorunlana, Ph.D.

Academic Editor

PLOS ONE

Journal Requirements:

Reviewers' comments:

Reviewer's Responses to Questions

**Comments to the Author**

1. Is the manuscript technically sound, and do the data support the conclusions?

Reviewer #1: Yes

Reviewer #2: Yes

2. Has the statistical analysis been performed appropriately and rigorously? 

Reviewer #1: N/A

Reviewer #2: N/A

3. Have the authors made all data underlying the findings in their manuscript fully available?

Reviewer #1: No

Reviewer #2: Yes

4. Is the manuscript presented in an intelligible fashion and written in standard English?

Reviewer #1: Yes

Reviewer #2: Yes

5. Review Comments to the Author

Reviewer #1: This is a very thorough, well-executed and thoughtful study.

General

•This is simply too long. Without tables it’s 10593 words for a qualitative survey of 14 people. No one is going to read that. The authors are strongly advised to cut the text length in half.

•The authors could reduce some length and ease the reader through making use of an additional graphic and table. See below.

Abstract:

•Methods:

oSurveys (how many?) and semi-structured interviews (12) in Zimbabwe regulators

oAlso interviewed (how many?) South African Health Products Regulatory Authority, Tanzania, and WHO to learn about “best practices in transitioning to medical device regulations”

oHow was survey developed, tested, how were respondents identified and survey distributed? How many sent? How many received? How were interviewees identified?

•Conclusion:

oAccessibility not the same concept as safety

Introduction

•Lines 40-46, are confusing and confused. These lines should be omitted, and making clear that the focus is on device manufacturer safeguards.

•Lines 72-74. I like how the authors describe overall regulatory deficiencies and that Zimbabwe is inclusive. I like how they review other literature exploring this. It would be quite helpful to precede this with an explanation of what the different Zimbabwean regulatory offices/bodies are, before delving into their deficiencies.

•Lines 75-76, the authors jump into describing stakeholder findings, before describing their research methods. This is out of place.

•Lines 80-83, the authors go from substandard and falsified medical devices, to manufacturer burden which are two separate concepts. The former is of primary responsibility, and the latter is of secondary interest. Further, I think a country with fewer regulations/oversight should not be equated with greater regulator burden for manufacturers.

•Line 87: I don’t think the authors are focused only on IVDs, are they?

Methods

•Line 103, Rugera requires a formal citation.

•How were respondents identified? How contacted? By whom?

•Line 113, “one staff member completed questionnaires…” goes in the results section

•Line 124: I don’t know what this means, as it is all about regulations.

•Line 142: how many representatives were selected?

•134-144, very confusing. Would help readers to draw a flow map or a table with all the different regulatory agencies (without and outside of Zimbabwe), and what their regulatory purview is, such as pre-market, post-market etc. (which may have to go in the results section but can be referenced here.). There are so many acronyms in the paper, it’s very tedious to read without a visual aid.

•184-185, who is the research team? How many designed, how many conducted, how many coded, how analyzed results?

Results

•Line 226, what’s the difference between a “participant “and a “critical stakeholder’

Discussion

•lines 604-606. This is the first mention of classification schema. Would help to introduce this earlier than the discussion section.

•Lines 694-707 should go into a table, along with other terms, to ease the readers.

•Lines 760-180 should go into a table.

•Line 814 “the main out of this study is a roadmap”. Where is it? Put it ino a visual, please. The text is too tedious.

Reviewer #2: General Comments

This relevant and timely study sheds light on the medical device regulatory landscape in a low-income country. The regulation of medical devices is critical to improving access to safe and reliable health products. The insights from key informants from the Medicines Control Authority, the Medical Laboratory and Clinical Scientists Council and the National Microbiology Reference Laboratory reveal both a lack of regulation as well as fragmented oversight involving these three agencies in Zimbabwe. This paper could potentially serve as a catalyst to improve engagement and communication between the three national institutions to create more robust and streamlined regulation of medical devices. The “best practice” perspectives from South Africa, Tanzania and the WHO highlight key enabling factors or facilitators towards developing a legislative framework for medical devices and for the successful implementation of medical device regulation.

The study design, sampling, data collection and data analysis methods are all suitable to address the research questions, although in some cases more detail is required to ensure transparency and reproducibility. Additionally, mechanisms for ensuring accuracy should be adequately explained.

The manuscript is generally well written, however, a number of language and typographical errors were found throughout the manuscript (e.g. line 157-159, line 165-166, line 240, line 496-498, line 816). The in-text citations should also be checked against the final list of references.

Introduction

It would be useful to provide an overview of the maturity levels of medicines regulatory authorities in African countries, including Zimbabwe to get a general sense of the regulatory capacity.

Line 62: What is meant by “no data”? Please clarify.

Line 82: Provide necessary references to support statements - unless the entire paragraph is entirely derived from reference 6?

Methods

It appears that no aspect of the methodology was piloted; is there a particular reason for this?

Line 99: Is there a particular reason why a legal expert from the medicines regulatory authority was not identified as a key informant? I imagine that the expert, technical knowledge of such a person would have been salient?

Line 103: Was the data collection tool validated for this setting? If not, an explanation of why this may not have been necessary should be included.

Line 113: It is not clear if one staff member from each of the three institutions was approached or whether all were approached and only three responded? If the former is correct, what was the rationale for including only one staff member from each agency and what was the basis for selecting these individuals? Were there any efforts to triangulate findings if only one participant was included from each institution? Were the questionnaires self-administered or researcher-administered? Also, were the surveys completed electronically, telephonically or in-person?

Line 100: Technically, the sampling approach combined a purposive and snowball method, correct? This should be clearly reflected.

Line 130: Is there a particular reason why the “GBT+Medical Devices” was not employed as the standard from which to develop the data collection tool? Please check the two references for the GBT as they appear to be incorrect.

Line 134-135: These acronyms were introduced previously and there is no need to write them out in full again.

Line 162: Is there a reference to support the medical device advancements in SA and Tanzania?

Table 1: Refer to comment for line 113. Additionally, were there only 12 participants who were approached for interviews or did some decline to be interviewed? There should be some indication of the response rate for the survey and for the interviews in paragraph where reference is made to Table 1.

Line 185: Were the same researchers involved in interviews and transcribing? If not, what mechanisms were introduced to ensure accuracy of the transcripts?

Line 191: Which coding frame was used for deductive coding? Elaborate on the manner in which inductive and deductive coding was used in tandem. This will enhance the reproducibility of the coding process.

Line 193: The coding was not done in duplicate, correct? What then was the process for achieving consensus?

Line 203: It makes sense to keep the interviewee’s identity anonymous but why/how was the “interview content” kept anonymous?

Results

The configuration of the results should be reconsidered since there appears to be repetition of the same quotes – sometimes up to three times in results (table and text) and discussion (text). Consider summarising findings in the table and only presenting quotes in the results.

P12: The theme of limited regulatory capacity fails to mention essential capacity issues that would generally plague a low-income country such as lack of technical expertise, infrastructure and funding. These challenges feature under a different theme , namely, the “improvements to the system” theme on P15, but should also feature under capacity constraints. Again, if the findings in the table are summarised, as opposed to presenting selected quotes, this could provide a more complete picture and prevent repetition.

P23-24: Formatting corrections are required in the table to align quotes with correct the correct interviewee.

Discussion

Line 597: Clarify the type of dossiers referred to in this sentence. Is it for condoms and gloves?

Line 694-708: It is important to reconcile this theory to the study findings in a short preface to this paragraph.

Line 716-717: Italicise for consistency

Line 740: Was this “study” also initiated by the Tanzanian NRA? If so, perhaps make this explicit.

Figure 1: Provide a citation for this figure

Figure 2: Is the roadmap novel in any way or is it adopted entirely from literature? If the latter, there should be a reference provided for the figure.

Overall, the paper's insights into the regulatory challenges and stakeholder perceptions in a resource-limited setting have the potential to resonate with a wide range of individuals and organizations involved in healthcare, policy, research, and advocacy on a global scale. Addressing these weaknesses alluded to above could enhance the overall quality, rigor, and clarity of the report, making it more robust and impactful in contributing to the understanding of medical device regulation in resource-limited settings like Zimbabwe.

Thank you for the opportunity to review this paper.

6. PLOS authors have the option to publish the peer review history of their article (what does this mean?). If published, this will include your full peer review and any attached files.

Reviewer #1: No

Reviewer #2: No

---

## [Author Response · Author response to Decision Letter 0]

16 Sep 2023

The response to the reviewers has more than 20,000 characters. Therefore, it has been uploaded as a separate attachment.

---

## [Decision Letter · Decision Letter 1]

19 Nov 2023

PONE-D-23-17224R1Regulation of medical devices in zimbabwe: a qualitative study with key stakeholdersPLOS ONE

Dear Dr. Chiku,

Thank you for submitting your manuscript to PLOS ONE. After careful consideration, we feel that it has merit but does not fully meet PLOS ONE’s publication criteria as it currently stands. Therefore, we invite you to submit a revised version of the manuscript that addresses the points raised during the review process.

We look forward to receiving your revised manuscript.

Kind regards,

Adetayo Olorunlana, Ph.D.

Academic Editor

PLOS ONE

Reviewers' comments:

Reviewer's Responses to Questions

**Comments to the Author**

1. If the authors have adequately addressed your comments raised in a previous round of review and you feel that this manuscript is now acceptable for publication, you may indicate that here to bypass the “Comments to the Author” section, enter your conflict of interest statement in the “Confidential to Editor” section, and submit your "Accept" recommendation.

Reviewer #3: (No Response)

2. Is the manuscript technically sound, and do the data support the conclusions?

Reviewer #3: Partly

3. Has the statistical analysis been performed appropriately and rigorously? 

Reviewer #3: I Don't Know

4. Have the authors made all data underlying the findings in their manuscript fully available?

Reviewer #3: No

5. Is the manuscript presented in an intelligible fashion and written in standard English?

Reviewer #3: No

6. Review Comments to the Author

Reviewer #3: This is an important paper but does require some major revisions.

overall the paper should be carefully read by all co-authors and someone not listed as an author for copy editing and improved writing style.

Introduction:

- line 114- what is 'this advice' given to Zimbabwe?

Methods:

- line 130- yes/no questionnaire needs more explanation. is it to understand opinions or is about existing structures/systems?

- line 134- interviews seem to be 2 types- those with KIIs in Zimbabwe and those outside Zimbabwe. Were different interview guides used for each type of KII? What was the purpose of each and what were the broad themes/topics covered for each? Results seem to indicate that interviews in Zimbabwe were to understand the current context, experiences, challenges, opportunities etc while those internationally were to understand best practices, frameworks, recommendations. The methods need to be clear to understand and align with the results.

- line 166- suggested rewording- "Purposive and snowball sampling was used to select participants, ... "

- line 205- one does not transcribe TO identify themes and concepts. These are 2 seperate steps that are not linked.

- line 206- reference for Rugera et al

- line 207-210- reword this to reflect the process of analysis and synthesis of qualitative data

- line 213- "Participants were assigned anonymous codes ... " suggest using IDs instead of codes to not confuse with coding of data. And this sentence should be moved to the 'ethical considerations' section.

- line 219- HOW were identities protected?

- Ethical considerations section- reorder, reword for improved readability

- line 211- 3 themes mentioned are not aligning with how themes/results presented in results section. There seems to be 3 different analyses done- describe each separately (1) yes/no questionnaire, (2) KIIs within Zimbabwe, (3) KIIs outside Zimbabwe. Was Rugger's frame used for both types of interviews

Results:

- line 224- results and discussion are 2 very seperate broad sections.

- include a few lines before presenting results to describe how the results are being presented, (1) yes/no questionnaire, (2) KIIs within Zimbabwe, (3) KIIs outside Zimbabwe.

- Present the Y/N questionnaire results in a table

- Table 2-

Themes listed here to not align with themes described in line 211- acts, regulations and guidelines.

The way the theme is named is presenting the result (ie. deleted limited and just have legal framework and regulatory capacity; delete 'the needs to improve the' and just leave it as medical devices regulatory system.Correct this in the subheadings of themes as well as in the table.

Delete 'Stakeholders agree that'

- lines 259-264- restructure for improved readability

- line 268- reword "At the time of the interviews, ... "

- line 278-279- the quote is not adding anything more than what is said at line 275

- section on 'medical devices regulatory system' needs descriptive text to summary the findings on this theme and not just rely on quotes. First quote on line 305 needs the pt ID included.

- Table 4- are these themes or seemingly more of recommendations which are useful but suggest they be listed in a table of recommendations or summary of findings

- headings at lines 336 and 344- combine into one subheading of 'international standards/tools'

- line 358- correct rep to be representative

Discussion:

- there are alot of results presented in the discussion. This needs to be reviewed to restructure so that it is discussing rather than presenting results and in particular quotes (data) should not be in a discussion. Many are good quotes and better presented in the results. To improve the structure of the discussion, consider subheadings.

- references are missing at lines 486/487, 491

- lines 403-412- this section repeats itself (methodological difference) and would benefit from some editing to improve readability

- lines 439- 441-

- Table 5 has important elements to consider but rather than laying out the table from Systems Thinking in Tobacco control, apply this to how it is or can be used in a system for regulation of medical devices in Zimbabwe. This has been done in Figure 2. Focus on describing figure 2 and table 5 is not necessary.

- lines 575-580- line 578 is repeating line 576. could this roadmap be introduced earlier in discussion to describe the challenges and opportunities based on this roadmap

- line 593- is data saturation necessary in a study of KIs/stakeholders describing the current context of a legal framework?

Conclusion:

-edit to improve readability. Currently reads like a bullet point list

7. PLOS authors have the option to publish the peer review history of their article (what does this mean?). If published, this will include your full peer review and any attached files.

Reviewer #3: No

---

## [Author Response · Author response to Decision Letter 1]

26 Nov 2023

Reviewers Comment

Methods:

- line 130- yes/no questionnaire needs more explanation. is it to understand opinions, or is about existing structures/systems?

Response: 

The aim of surveying stakeholders in Zimbabwe was to comprehend medical device regulations in Zimbabwe and understand the current context, structures, experiences, challenges, and opportunities. This has resulted in the amendment of the statement: "From June to November 2022, using questionnaires and interviews, surveyed stakeholders to comprehend medical device regulations in Zimbabwe to understand the current context, experiences, challenges, and opportunities.” (lines 131-135). 

Reviewer’s comment: 

- line 134- interviews seem to be 2 types- those with KIIs in Zimbabwe and those outside Zimbabwe. Were different interview guides used for each type of KII? What was the purpose of each and what were the broad themes/topics covered for each? Results seem to indicate that interviews in Zimbabwe were to understand the current context, experiences, challenges, opportunities etc while those internationally were to understand best practices, frameworks, recommendations. The methods need to be clear to understand and align with the results.

Response: 

The aim of surveying stakeholders in Zimbabwe was to comprehend medical device regulations in Zimbabwe to understand the current context, structures, experiences, challenges, and opportunities. This has resulted in the amendment of the statement: "From June to November 2022, using questionnaires and interviews, surveyed stakeholders to comprehend medical device regulations in Zimbabwe to understand the current context, experiences, challenges, and opportunities.” (Lines 131-135). 

The explanation for interviewing regulators from South Africa, Tanzania and the World Health Organization was provided as shown below: 

“Furthermore, to understand best practices, frameworks, and recommendations for establishing and improving medical device regulations from other national regulatory authorities in the SADC region, semi-structured interviews were conducted with representatives from the South African Health Products Regulatory Authority (SAHPRA) and one from the Tanzania Medicines and Medical Devices Authority (TMDA). Additionally, a WHO’s Regulatory Systems Strengthening regulator was interviewed to gain perspective on using the Global Model Regulatory Framework for Medical Devices (GMRFMD) and how member states have been using it to establish and strengthen their medical device regulatory systems.” (lines 172-179).

Reviewer’s comment: 

- line 166- suggested rewording- "Purposive and snowball sampling was used to select participants, ... "

Response: The statement was reworded: "Purposive and snowball sampling was used to select participants, including representatives from medical device regulation institutions.” (lines 186-187). 

Reviewer’s comment: 

- line 206- reference for Rugera et al.

Response: The citation and reference for Rugera et al was added in line 229, “ Data were deductively coded using Rugera et al.'s frame and themes, with inductive codes added (14).”

Reviewer’s comment: 

- line 207-210- reword this to reflect the process of analysis and synthesis of qualitative data.

Response: The section was reworded as follows, “Data were deductively analyzed and synthesized using Rugera et al.'s frame and themes, with inductive codes added (14) following an iterative process that involved making sense of the rich, non-numeric information gathered during qualitative research. To achieve this, the following steps were taken: 

• Familiarization with the data involved transcribing information and immersing ourselves in its content.

• Coding of the data encompassed open coding, axial coding, and selective coding.

• Theme development comprised grouping codes into primary themes and breaking them into sub-themes.

• The constant comparison method involved ongoing scrutiny of the data to validate emerging patterns. Throughout this process, we documented our thoughts and reflections to facilitate the development of concepts and theories.

• Member checking, a validation step, entailed sharing our findings with participants to ensure that our interpretations aligned with their experiences and perspectives.

• Synthesis involved integrating individual themes and concepts to comprehensively understand medical device regulation in Zimbabwe and the broader region. This step aimed to construct a coherent narrative of the findings.

• Reporting was the final step, focusing on clearly articulating the findings. We utilized quotes and examples to illustrate key points and reflected on biases and their potential influence on the analysis.

CC and FB were involved in the synthesis and analysis process of the data.”(lines 228-247). 

Reviewer’s comment: 

- line 213- "Participants were assigned anonymous codes ... " suggest using IDs instead of codes to not confuse with coding of data. And this sentence should be moved to the 'ethical considerations' section.

Response: As suggested, " codes " were replaced with anonymous identifiers. Furthermore, the statement was moved to the ethical considerations section: "The consent form guaranteed anonymity, and real names were omitted from reports and publications. Participants were given anonymous identifiers based on their organizations (e.g., MCAZ KI07 for an MCAZ participant). Interviews were conducted anonymously, and these identifiers were used for analysis and reporting.” (lines 251-254). 

Reviewer’s comment: 

- line 219- HOW were identities protected?

Response: Identities were protected by data de-identification of the respondents by anonymizing by removing names and job titles of the participants. The statement has been edited as follows, “The consent form guaranteed anonymity, and real names were omitted from reports and publications. Participants were given anonymous identifiers based on their organizations (e.g., MCAZ KI07 for an MCAZ participant). Interviews were conducted anonymously, and these identifiers were used for analysis and reporting.” (lines 259-261). 

Reviewer’s comment

- Ethical considerations section- reorder, reword for improved readability.

Response: The ethical considerations section was improved as follows, “The study (MRCZ/A/2900) received approval from the Medical Research Council of Zimbabwe. Participants had the option to withdraw at any time by providing written consent. During interviews, confidentiality was ensured, and identities were safeguarded. The consent form guaranteed anonymity, and real names were omitted from reports and publications. Participants were given anonymous identifiers based on their organizations (e.g., MCAZ KII7 for an MCAZ participant). Interviews were conducted anonymously, and these identifiers were used for analysis and reporting.

To maintain confidentiality, interviews and documents were securely stored with restricted access. Digital files were both password-protected and encrypted. This safeguarded participant identities by anonymizing identifiers and removing names and job titles from the data.”(lines 251-261). 

Reviewer’s Comment: 

- line 211- 3 themes mentioned are not aligning with how themes/results presented in results section. There seems to be 3 different analyses done- describe each separately (1) yes/no questionnaire, (2) KIIs within Zimbabwe, (3) KIIs outside Zimbabwe. Was Rugger's frame used for both types of interviews? 

Response: Thank you for raising this comment. This section was not very clear. Rugera’s framework was only used for the interviews conducted in Zimbabwe. This section has been moved to the results section since it was not in the right place in the previous version. 

Reviewer’s comment:

Results:

- line 224- results and discussion are 2 very separate broad sections.

Response: Results and discussion were separated. The discussion was deleted, as shown in line 262. 

Reviewer’s Comment: 

- include a few lines before presenting results to describe how the results are being presented, (1) yes/no questionnaire, (2) KIIs within Zimbabwe, (3) KIIs outside Zimbabwe.

- Present the Y/N questionnaire results in a table.

Response: The introduction to the results section and how results will be presented was added as follows, “We present the study's findings that aimed to examine the medical device regulatory landscape in Zimbabwe and lessons learned from SAHPRA, TMDA, and WHO RSS in establishing and strengthening medical device regulations. The results are organized into three subsections, each addressing specific research questions. The first sub-section presents results from the questionnaires, the second from the semi-structured interviews of stakeholders in Zimbabwe, and the third from interviewing regulators from SAHPRA, TMDA and WHO RSS.” (lines 263-268). 

Reviewer’s comment

- Table 2-

Themes listed here to not align with themes described in line 211- acts, regulations and guidelines.

The way the theme is named is presenting the result (ie. deleted limited and just have legal framework and regulatory capacity; delete 'the needs to improve the' and just leave it as medical devices regulatory system.Correct this in the subheadings of themes as well as in the table.

Response: Themes in line 211 of the previous manuscript were deleted because they were not meant to be there. The themes have been corrected in lines 297-379 and in table 2 as shown below, 

“Twelve participants were interviewed, yielding a 92.3% response rate (one refusal). Data were categorized into regulatory system components, revealing six main themes after transcript analysis: legal framework, regulatory capacity, conformity assessment, post-market surveillance, medical device regulation, and collaboration of regulatory authority.”

Table 2: Primary Themes Arising from Interviews with Regulatory Stakeholders in the Medical Device Sector in Zimbabwe: A Study Spanning June to November 2022.

Theme Summary Findings 

Legal framework The MCAZ lacked a legal basis to regulate medical devices. The MLCScCZ attempted to use the Health Professions Act to regulate IVDs, but it was not applicable.

Regulatory capacity The MCAZ and MLCScCZ lacked technical expertise, funding, and infrastructure to regulate medical devices, including IVDs. The NMRL lacked technical expertise, competencies, and reference materials to verify IVD safety and performance, which the MLCScCZ was responsible for.

Conformity assessment Glove and condom conformity assessment was identical despite different risk classes. 

Post-market surveillance No post-market surveillance is needed for public safety. 

Medical devices regulatory system Stakeholders agreed that a balanced, risk-based IVD regulatory system must be established and strengthened. Training and technical support, infrastructure, trained personnel, and transparency and accountability mechanisms are needed. 

Collaboration of regulatory authorities MCAZ, MLCScCZ, NMRL and other stakeholders must collaborate using systems thinking to ensure IVD regulations are risk-based, responsive, and effective. 

Note. MCAZ = Medicines Control Authority of Zimbabwe; MLCScCZ = Medical Laboratory and Clinical Scientists Council of Zimbabwe; NMRL = National Microbiology Reference Laboratory.

Reviewer’s comment 

Delete 'Stakeholders agree that'

Response: Deleted as suggested. 

Reviewer's comment

- lines 259-264- restructure for improved readability. 

The statement was rephrased as, “ The Health Professions Act, employed by MLCScCZ, lacks provisions pertaining to the oversight of In Vitro Diagnostic Devices (IVDS). Although the Medicines Control Authority of Zimbabwe (MCAZ) has drafted import/export regulations for medical devices, awaiting approval, it has also expressed intentions to amend the Medicines and Allied Substances Control Act (MASCA) to encompass the regulation of all health products. MCAZ contemplates a name change to reflect a broader scope beyond medicines, a matter that has been discussed within the Ministry of Health. The regulatory framework for In Vitro Diagnostic Devices (IVDs) under the National Microbiology Reference Laboratory (NMRL) remains ambiguous, with an interviewee acknowledging the difficulty in locating the specific Act of Parliament delineating NMRL's IVD regulatory functions (NMRL KI08)." (lines 317-324). 

Reviewer’s comment

- line 268- reword "At the time of the interviews, ... "

Response: Reworded as suggested, “ At the time of the interviews, the MCAZ only regulated condoms and gloves.” (line 328). 

Reviewer’s comment

- line 278-279- the quote is not adding anything more than what is said at line 275.

Response: The quote was deleted. 

Reviewer’s comment

- section on 'medical devices regulatory system' needs descriptive text to summarise this theme's findings and not just rely on quotes. The first quote on line 305 needs the pt ID included.

Response: A descriptive text was added and read as, “ It was found that the medical devices regulatory system in Zimbabwe was not proactive, effective, or proportional to the risk class of medical devices. However, it is reactive to the disease burden if the disease is programmatically significant but without a proper regulatory framework based on legal provisions.” (lines 365-368)

The participant ID for the participant who commented was added as MLCScCZ KI12. (line 371). 

Reviewer’s comment: 

- Table 4- are these themes or seemingly more of recommendations which are useful but suggest they be listed in a table of recommendations or summary of findings?

Response: These are the best practices and lessons learnt from regulators outside of Zimbabwe. An introduction has been included to explain the process of analysis and generation of recommendations, “ WHO, SAHPRA and TMDA regulators were interviewed to learn best practices that can be used to make recommendations on the medical devices roadmap for Zimbabwe. Best practices obtained through interviews are typically reported as recommendations. Qualitative analysis of the data from interviews was used to generate themes representing patterns or recurring topics that emerge from the data. These themes were then used to inform the generation of recommendations. Insights were gained from these interviews, and interviewees' characteristics and themes are summarized in Tables 3 and 4.” (lines 391-395). 

Reviewer’s comment

- headings lines 336 and 344- combine into one subheading of 'international standards/tools'

Response: Headings below line 412 and line 419 are now combined under international standards and tools. 

Reviewer’s comment

- line 358- correct rep to be representative.

Response: Rep was replaced with representative, as suggested by the reviewer in line 436 of the manuscript. 

Reviewer’s comment:

Discussion:

- there are alot of results presented in the discussion. This needs to be reviewed to restructure so that it is discussing rather than presenting results and in particular quotes (data) should not be in a discussion. Many are good quotes and better presented in the results. To improve the structure of the discussion, consider subheadings.

Response: Quotes were removed from the discussion. Sub-headings were included in the discussion. The subheadings are Current Regulatory Landscape in Zimbabwe and Exploring Opportunities for Improvement. 

Reviewer’s comment

- references are missing at lines 486/487, 491.

Response: The citation and references were added in lines 555-557, “Dube-Mwedzi et al. and Stanislav et al. identified inadequate capacity and effort in post-market surveillance, prevention, detection, and response to substandard and falsified medical products (12,25).”

Reviewer’s comment

- lines 403-412- this section repeats itself (methodological difference) and would benefit from some editing to improve readability.

Response: This section was rephrased to improve readability as follows, “Investigation of the regulatory landscape in Zimbabwe is meant to develop a roadmap for a medical device regulatory system to address the challenges and leverage the opportunities identified. The importance of medical device regulations upholds safety, quality, and performance standards for public health (8). Effective regula

---

## [Decision Letter · Decision Letter 2]

18 Dec 2023

PONE-D-23-17224R2Regulation of medical devices in Zimbabwe: a qualitative study with key stakeholdersPLOS ONE

Dear Dr. Chiku,

Thank you for submitting your manuscript to PLOS ONE. After careful consideration, we feel that it has merit but does not fully meet PLOS ONE’s publication criteria as it currently stands. Therefore, we invite you to submit a revised version of the manuscript that addresses the points raised during the review process.

We look forward to receiving your revised manuscript.

Kind regards,

Adetayo Olorunlana, Ph.D.

Academic Editor

PLOS ONE

Reviewers' comments:

Reviewer's Responses to Questions

**Comments to the Author**

1. If the authors have adequately addressed your comments raised in a previous round of review and you feel that this manuscript is now acceptable for publication, you may indicate that here to bypass the “Comments to the Author” section, enter your conflict of interest statement in the “Confidential to Editor” section, and submit your "Accept" recommendation.

Reviewer #3: (No Response)

2. Is the manuscript technically sound, and do the data support the conclusions?

Reviewer #3: Partly

3. Has the statistical analysis been performed appropriately and rigorously? 

Reviewer #3: N/A

4. Have the authors made all data underlying the findings in their manuscript fully available?

Reviewer #3: No

5. Is the manuscript presented in an intelligible fashion and written in standard English?

Reviewer #3: No

6. Review Comments to the Author

Reviewer #3: (line numbers refer to version with track changes)

Abstract

- Intro is 2 sentences. Need an additional 1-2 sentence to give background.

- It needs to be clear that different methods employed- a questionnaire and KIIs.

- Analysis of questionnaires in main paper is descriptive analysis but not reflected in abstract.

- Abstract is too long and needs revision- consider condensing the conclusion as a start.

Intro

- Line 139- who recommended member states in the SADC region...?

- Line 169- how was it sent back to researcher? “... and emailed back to the researcher”?

- Line 170- “study’s standardize scope prevented triangulation”- belongs in limitations section

Methods

- Line 208-211. Inset that these are based in Zimbabwe to make it clear.

- Line 216- separate paragraph when talking about interviews done outside Zimbabwe

- Line 240- move to study procedures in first section of methods. How were data recorded? Audio recordings, notes... ?

- Lines 252-244 are not part of analysis. This is describing study procedures and should be moved to that section of the methods.

- Line 258-9- transcription and identifying themes and concepts are not interrelated. Move transcription to how data collected and processed. Then lines 257-260 should be restructured for improved readability and avoid repetition.

- Line 267-271 talks about development of themes but its stated that themes come from Rugera’s themes. This needs to be clarified.

Findings

- Survey results- A table of what exists and does not exist in the regulatory bodies would be beneficial.

- Table 2- no need to include the study duration in the title

- Line 436- italicize the quote

- Line 459 saying the same thing as previous sentence.

- Line 460-462 should be moved to analysis section of methods.

- Table 4- no need to include dates in the title

- Line 481- ‘tolls’- should this be ‘tools’?

- Line 510- is this the first time these 3 acronyms used?- need to write them out in full.

- Line 573, “despite...” this sentence is incomplete. Rephrase.

- Line 589- what is meant by ‘work’? be more specific.

- Line 638-9- rewrite for improved readability

- Line 657- “this finding is similar...” this can be merged into previous sentence to for improved readability.

- Lines 659 and 664 can be rewritten to merge and improve readability.

- Line 712- its not clear what this caption credit is for. Is it a reference for the requirements for effective regulation? If so cite it as any other reference.

- Line 734- “Table 6...” reword to be more clear. “...indicates gaps in MASCA identified from the South Africa and Tanzania experiences.”

- Line 742-755 is giving a lot of results rather then discussing.

- Line 773- “This framework covers...”- remove this. It’s a repeat of line 770

Conclusion

- This is very long. Lines 797-836- are these another summary of what is in the discussion? Suggest to remove this while assuring that its already covered in the results/discussion.

Double check that ALL acronyms are described in full the first time they are used

This is valuable research and worthy of publication. But I strongly recommend the manuscript be read and edited by someone else to check its well organized and written clearly.

7. PLOS authors have the option to publish the peer review history of their article (what does this mean?). If published, this will include your full peer review and any attached files.

Reviewer #3: No

---

## [Author Response · Author response to Decision Letter 2]

1 Feb 2024

Reviewer’s comment

Abstract

- Intro is 2 sentences. Need an additional 1-2 sentence to give background.

Response: 

The abstract was reviewed to give the background of the study as follows, “However, there is a disparity in the availability of good quality and high-performing products due to the delay in the readiness of medical devices and IVD regulations in Low and Middle-Income Countries (LMICs), especially when compared to the progress made globally over the past 20 years global average.” (lines 36-39). 

Reviewer’s comment

- It needs to be clear that different methods employed- a questionnaire and KIIs.

The statement was reviewed to state the methods that were used for the various participants as follows, “Between June and November 2022, we used purposive and snowball sampling techniques to select three participants from the Medicines Control Authority of Zimbabwe (MCAZ), the Medical Laboratory and Clinical Scientists Council of Zimbabwe (MLCScCZ), and the National Microbiology Reference Laboratory (NMRL) who completed the questionnaire. Twelve participants represented regulators from these institutions to understand the current status of medical device and IVD regulations and their relationships. Key informant interviews were conducted with three regulators from SAHPRA, TMDA, and the WHO RSS to learn the best practices for transitioning to medical device and IVD regulations to develop a roadmap for Zimbabwe.” (lines 44-57). 

Reviewer’s comment:

- Analysis of questionnaires in main paper is descriptive analysis but not reflected in abstract.

Response: 

Analysis in the abstract was reviewed and reads as follows, “We used descriptive analysis to analyze and examine data from questionnaires. Thematic approach and inductively developed a common coding framework to analyze emerging themes from interviews conducted in Zimbabwe and to generate recommendations from interviewing regulators from SAHPRA, TMDA, and WHO RSS” (lines 57-58). 

Reviewer’s comment

- Abstract is too long and needs revision- consider condensing the conclusion as a start.

Response: 

The conclusion in the abstract was condensed as follows, “The findings reveal significant deficiencies and gaps in the legal framework for regulating medical devices and IVDs, particularly highlighting the need for a legal framework and the absence of more comprehensive regulations. Regulatory entities like MCAZ face capacity limitations, especially in regulating medical devices and IVDs. Conformity assessment processes, medical device and IVD classification criteria, and post-market surveillance also represent challenges, highlighting the need for a well-defined framework and regulatory procedures. The Zimbabwean regulatory system pathway is characterized as reactive, prompting a collection of regulations on initiatives to address needs. Despite facing challenges, there is a recognition of the importance of collaboration among regulatory authorities, emphasizing a shared commitment to improving and strengthening medical device and IVD regulations for improved patient safety. By advocating for a proactive, comprehensive, and legally sound approach, indicating the potential for collaborations and synergies, the study provides a foundation for well-informed policy recommendations to guide enhancements and build a framework for a resilient, efficient and transparent regulatory environment in Zimbabwe and African region as a whole.” (lines 61-75). The abstract is still within the required 500 words limit. 

Reviewer’s comment

Intro

- Line 139- who recommended member states in the SADC region...?

Response: The statement and citation were removed to reduce the introduction to literature that is specific for Zimbabwe. 

Reviewer’s comment

- Line 169- how was it sent back to researcher? “... and emailed back to the researcher”?

The questionnaires were sent to the researcher by email. The statement was rephrased to read as follows, “The questionnaire was self-administered electronically and sent to the researcher by email.” (line 149-150). 

Reviewer’s comment

- Line 170- “study’s standardize scope prevented triangulation”- belongs in limitations section

Response: 

The lines was moved to the limitations section (line 688-689). 

Reviewer’s comment

Methods

- Line 208-211. Inset that these are based in Zimbabwe to make it clear.

Response: These three institutions are based in Zimbabwe. (lines 212-213-195)

Reviewer’s comment

- Line 216- separate paragraph when talking about interviews done outside Zimbabwe.

Response: 

A separate paragraph was created as suggested; please refer to line 220. 

Reviewer’s comment

- Line 240- move to study procedures in first section of methods. How were data recorded? Audio recordings, notes... ?

Response:

Moved to procedures in lines 203-205 and reads as follows, “Zoom and Teams were used for remote interviews. After consent, the data from interviews were recorded through audio recording.”

Reviewer’s comment

- Lines 252-244 are not part of analysis. This is describing study procedures and should be moved to that section of the methods.

Response: 

The section was moved to the methods section in lines 201-203, “ Researcher (CC) introduced himself during interviews to participants, disclosing identity, credentials, occupations, gender, experiences, and training. To prevent bias, he avoided contact with participants before the interviews.”

Reviewer’s comment

- Line 258-9- transcription and identifying themes and concepts are not interrelated. Move transcription to how data collected and processed. Then lines 257-260 should be restructured for improved readability and avoid repetition.

Response: The concepts were separated. Transcription was moved to the methods section. The analysis was improved for readability from lines 252-267, “Analysis of data from key informant interviews used a theoretical content analysis framework and NVivo 12. The analysis process consisted of deductive analysis supplemented by inductive codes in an iterative process. Key steps in this process included familiarization with the data, achieved through transcribing information and immersing in its content. Coding of the data encompassed open coding, axial coding, and selective coding. Theme development involved grouping codes into primary themes and breaking them into sub-themes.

The constant comparison method was applied, entailing ongoing scrutiny of the data to confirm emerging patterns. Thoughts and reflections were documented throughout this process to develop concepts and theories. Member checking, a validation step, involved sharing findings with participants to align interpretations with their experiences. Synthesis aimed at integrating individual themes and concepts to comprehensively understand medical device regulation in Zimbabwe and the broader region, constructing a coherent narrative of the findings.

The final step, reporting, focused on clearly articulating the findings. The author utilized quotes and examples to illustrate key points, while also reflecting on biases and their potential influence on the analysis. CC and FB were involved in the synthesis and analysis process of the data. ”

Revirewer’s comment: 

- Line 267-271 talks about development of themes but its stated that themes come from Rugera’s themes. This needs to be clarified.

Response: The line was clarified. It reads as follows, “Analysis of data from key informant interviews used a theoretical content analysis framework and NVivo 12. The analysis process consisted of deductive analysis supplemented by inductive codes in an iterative process.” (lines 252-257). 

Reviewer’s comment

Findings

- Survey results- A table of what exists and does not exist in the regulatory bodies would be beneficial.

Response:

The table was included, comparing what exists and does not exist based on the WHO and IMDRF frameworks. See lines 313-315. 

Reviewer’s comment

- Table 2- no need to include the study duration in the title

Response: 

The study duration was removed. See line 345. 

Reviewer’s comment:

- Line 436- italicize the quote

Response: the quote was italised, see lines 605-606. 

Reviewer’s comment

- Line 459 saying the same thing as previous sentence.

Response: The repeated line was deleted. See line 436. 

Reviewer’s comment

- Line 460-462 should be moved to analysis section of methods.

Response: The section was deleted as it was not adding value to the data analysis section. See lines 435-439. 

Reviewer’s comment

- Table 4- no need to include dates in the title

Response: Dates deleted from the title (see line 433). 

Reviewer’s comment

- Line 481- ‘tolls’- should this be ‘tools’?

Response: My apologies for the typographical error. It was corrected to read as tool/ (see line 604). 

Reviewer’s comment

- Line 510- is this the first time these 3 acronyms used?- need to write them out in full.

Response: The statement was rephrased as follows, “Tanzania aligned its regulations with the regulations from the United States of America Food and Drug Administration (USFDA), the Australian Therapeutic Goods Administration (TGA)studied USFDA, TGA, and the Japanese Pharmaceutical and Medical Devices Agency(PMDA).” (lines 487-490). 

Reviewer’s comment

- Line 573, “despite...” this sentence is incomplete. Rephrase.

Response: The statement was reviewed and reads as, “Despite the MLCsCZ registration of regulations, HIV, Malaria, TB, Syphilis, and COVID-19 IVDs through performance evaluations, the approach is ineffective.” (lines 629-632). 

Reviewer’s comment

- Line 589- what is meant by ‘work’? be more specific.

Response: The statement was reviewed to read as, “ Authorities in medical device and IVD regulation play a pivotal role in shaping policies, ensuring effective oversight, fostering collaboration, adapting to technological changes, and prioritizing public health. Their decisions directly impact the safety and efficacy of medical devices, as well as the overall well-being of patients and the healthcare system.” (lines 692-695). 

Reviewer’s comment

- Line 638-9- rewrite for improved readability. 

Response: The statement was rephrased to read as follows, “Reliance and recognition mechanisms involve an NRA deciding on product approval, even if it relies on other authorities. The relying authority is independent and accountable for decisions. Regulatory reliance can reduce access barriers and use resources efficiently. It can reduce uncertainties for innovators and improve crisis responses. The approach speeds up access to safe and quality health technology.” (lines 729-734). 

Reviewer’s comment

- Line 657- “this finding is similar...” this can be merged into previous sentence to for improved readability.

Response: The statement was merged as requested, “Post-market surveillance documentation is lacking, with the MCAZ only monitoring condoms and gloves in the market. While the drafted regulations for medical device import and export and IVD include some post-market surveillance elements, they are not comprehensive. This aligns with the findings of Hubner et al. (10). Both Dube-Mwedzi et al. and Stanislav et al. have highlighted insufficient capacity and effort in post-market surveillance, specifically in the prevention, detection, and response to substandard and falsified medical products (24,25). Our study confirms a correlation with the findings of Stanislav and Mwedzi-Dube, though the studies were for the whole of Southern Africa, Zimbabwe included.” (lines 789-800). 

Reviewer’s comment

- Lines 659 and 664 can be rewritten to merge and improve readability.

Response: Please refer to the response above.

Reviewer’s comment:

- Line 712- its not clear what this caption credit is for. Is it a reference for the requirements for effective regulation? If so cite it as any other reference.

Response: The caption was adapted to medical device regulations. It was originally from the Tobacco Control Monograph. The caption was deleted and the citation was added in line 840. 

Reviewer’s comment: 

- Line 734- “Table 6...” reword to be more clear. “...indicates gaps in MASCA identified from the South Africa and Tanzania experiences.”

Response: The statement was rephrased to read as follows, “ Table 6 shows changes required in Zimbabwe's Medicines and Allied Substances Control Act based on the best practices learned from delineates divergences derived from the experiences in South Africa and Tanzania. , offering insights substantiating the proposed modifications to Zimbabwe's Medicines and Allied Substances Control Act (MASCA)..” (lines 865-869). 

Reviewer’s comment: 

- Line 742-755 is giving a lot of results rather then discussing.

Response: We agree with the reviewer’s comment. This section was deleted. 

Reviewer’s comment: 

- Line 773- “This framework covers...”- remove this. It’s a repeat of line 777.

Response: This section was deleted; see lines 841-842. 

Reviewer’s comment

Conclusion

- This is very long. Lines 797-836- are these another summary of what is in the discussion? Suggest to remove this while assuring that its already covered in the results/discussion.

Response: We agree with the reviewer’s comment. Lines 794-840 were deleted.

---

## [Decision Letter · Decision Letter 3]

26 Feb 2024

PONE-D-23-17224R3Navigating Regulatory Landscape: A Qualitative Exploration of Medical Devices and In Vitro Diagnostic Medical Devices Oversight in Zimbabwe through Key Stakeholder PerspectivesPLOS ONE

Dear Dr. Chiku,

Thank you for submitting your manuscript to PLOS ONE. After careful consideration, we feel that it has merit but does not fully meet PLOS ONE’s publication criteria as it currently stands. Therefore, we invite you to submit a revised version of the manuscript that addresses the points raised during the review process.

We look forward to receiving your revised manuscript.

Kind regards,

Adetayo Olorunlana, Ph.D.

Academic Editor

PLOS ONE

Journal Requirements:

Reviewers' comments:

Reviewer's Responses to Questions

**Comments to the Author**

1. If the authors have adequately addressed your comments raised in a previous round of review and you feel that this manuscript is now acceptable for publication, you may indicate that here to bypass the “Comments to the Author” section, enter your conflict of interest statement in the “Confidential to Editor” section, and submit your "Accept" recommendation.

Reviewer #3: (No Response)

Reviewer #4: (No Response)

2. Is the manuscript technically sound, and do the data support the conclusions?

Reviewer #3: Yes

Reviewer #4: Yes

3. Has the statistical analysis been performed appropriately and rigorously? 

Reviewer #3: Yes

Reviewer #4: I Don't Know

4. Have the authors made all data underlying the findings in their manuscript fully available?

Reviewer #3: Yes

Reviewer #4: Yes

5. Is the manuscript presented in an intelligible fashion and written in standard English?

Reviewer #3: (No Response)

Reviewer #4: Yes

6. Review Comments to the Author

Reviewer #3: Abstract- this is still very long and needs to be shortened

line 40- remove 'global average'

line 42- "opacity that is opaque"- this is not clear. consider rewording

line 43- "it is unclear if medical devices..." this is a repeat of previous sentence. suggest this sentence be removed

line 46- incomplete sentence. add, "Additionally we aimed, to learn..."

line 49- 3 pts from 3 organisations is 9 people. Line 53 says 12 pts. Please double check these numbers

line 58- "thematic approach" sentence needs to be restructured. "Using a common coding framework, we used thematic analysis for interviews conduced in Zimbabwe and with international regulators."

Methods:

Table 1 goes in results

line 255- NVivo does not analyse. It is used top code and organise data. Move mention of its use in line 258, "Coding of data... and selective coding, using NVivo 12."

line 276- you discuss anonymity yet the participants organisation and their job roles is listed. This is very identifying. In line 282 its say removing job titles from data. Yet job title is in tables 1 and 4. Please address this important ethical issue.

Results:

Table 2- shorten the questions to be key phrases/points

Discussion:

line 568- i believe this is first time RLS is used. write it out in full

in the section 'exploring opportunities for improvement' there are still results presented in the form of quotes. This discussion needs to be more tidy by summarising and discussing results rather than presenting them.

I recommend that authors have someone external to the project review this paper in detail. There is still some room for improvement in the writing style to be more concise and scientific.

Reviewer #4: Thank you for the opportunity to review this interesting article which looks at medical device regulations in Zimbabwe and Tanzania. The article includes recommendations for further maturity of regulations and regulatory structures in these countries, and potentially these are applicable more widely in similar settings.

Firstly, I would like to check with the authors that they have explicit consent for the details on job title used here and especially in table 4 this is quite an identifying characteristic (just short of naming participants in many cases).

I have a number of other, minor comments, with the hope to improve readability and therefore bring forward the main learning of this manuscript.

Introduction

Line 82: “It also includes products…” what is “it” referring to?

Line 104: please introduce the IMDRF acronym here and remove full name from line 109.

Line 114: I have not seen the African region referred to as the “Afro” region – is this common? Perhaps use “African region” instead

Line 119-120: Reference for this?

Methods

General – authors state that COREQ is used for reporting however several things are left out: what are the interviewers' experience and training? what is the relationship, if any, between the interviewer and the participants? Any prior knowledge?

Line 143-145: The sentence that begins with “However, our study…” belongs in limitations section. Also I don’t fully understand this, please rephrase for clarity.

Medical devices and IVD regulatory system

Line 392 – 394 – regarding the sentence that begins “However, it is reactive…” I see that the further text below is an example of what you mean by this so perhaps you don't need this sentence? If you do decide to keep it, please rephrase to add clarity.

International standards and tools

Line 442 – On first reading I was wondering what the participant mean that the model was updated? Now I see there is a section currently in the discussion (Line 630 – 659 and table 5), which if it is indeed part of what you found in the study, is results and should be moved up here specifically as it helps to explain this paragraph and statement by the participant.

The transition period to implement new regulations

Line 489: what does “rely on the status” mean?

Line 491-2: Please clarify whose statement is this - the authors of this manuscript or the participants. If this is the authors’ statement it belongs in the discussion.

Discussion

Line 630 – 659 and table 5 – I think this belongs in the results section.

Line 638 – “Table 6” is below labelled table 5.

7. PLOS authors have the option to publish the peer review history of their article (what does this mean?). If published, this will include your full peer review and any attached files.

Reviewer #3: No

Reviewer #4: **Yes: **Sebastian S Fuller

---

## [Author Response · Author response to Decision Letter 3]

6 Mar 2024

NOTE: The line numbers refer to the “ Manuscript with Tracked Changes” in Simple Matkup Mode of Tracked Changes. 

Reviewer #3: Abstract- this is still very long and needs to be shortened

Line 40- remove 'global average'

Response: “ Global average” was deleted. The abstract was reduced from 499 to 355 words. 

Reviewer’s comment 

line 42- "opacity that is opaque"- this is not clear. consider rewording

Response: Rephrased and reads, "Zimbabwe's regulatory framework for medical devices and IVDs is unclear, leading to ineffective compliance and surveillance. Our study aimed to explore the current status of medical devices and IVD regulations in Zimbabwe.” (lines 37-39). 

Reviewer’s comment

Line 43- "It is unclear if medical devices..." this is a repeat of the previous sentence. Suggest this sentence be removed.

Response: Deleted line 43. 

Reviewer’s comment 

Line 46- incomplete sentence. Add, "Additionally, we aimed to learn..."

Response, edited and reads in lines 41-50, “Semi-structured interviews were conducted with 12 regulators from the Medicines Control Authority of Zimbabwe (MCAZ) National Microbiology Reference Laboratory (NMRL), Medical Laboratory and Clinical Scientists Council (MLCScCZ) to understand the current status of medical devices and IVD regulations in Zimbabwe. Three participants completed a questionnaire to understand the regulatory landscape in Zimbabwe. Three key informant interviews were conducted with three regulators from the South African Health Products Regulatory Authority (SAHPRA), Tanzanian Medicines and Medical Devices Authority (TMDA), and World Health Organization Regulatory Systems Strengthening (WHO RSS) to learn best practices to create a roadmap for Zimbabwe.”

Reviewer’s comment

Line 49- 3 pts from 3 organizations is 9 people. Line 53 says 12 pts. Please double-check these numbers.

Response: The statement has been rephrased to correct the number of participants that were interviewed as follows, “Semi-structured interviews were conducted with 12 regulators from the Medicines Control Authority of Zimbabwe (MCAZ) National Microbiology Reference Laboratory (NMRL), Medical Laboratory and Clinical Scientists Council (MLCScCZ) to understand the current status of medical devices and IVD regulations in Zimbabwe. Three participants completed a questionnaire to understand the regulatory landscape in Zimbabwe. Three key informant interviews were conducted with three regulators from the South African Health Products Regulatory Authority (SAHPRA), Tanzanian Medicines and Medical Devices Authority (TMDA), and World Health Organization Regulatory Systems Strengthening (WHO RSS) to learn best practices to create a roadmap for Zimbabwe.” (lines 41-50)

Reviewer’s comment 

line 58- "thematic approach" sentence needs to be restructured. "Using a common coding framework, we used thematic analysis for interviews conducted in Zimbabwe and with international regulators."

Response: The statement has been amended to “We analyzed qualitative data using a thematic analysis” (Lines 50-51).

Reviewer’s comment

Methods:

Table 1 goes in results

Response: 

Table 1 was moved to the results section. (See lines lines 282-284). 

Reviewer’s comment

line 255- NVivo does not analyze. It is used top code and organize data. Move mention of its use in line 258, "Coding of data... and selective coding, using NVivo 12."

Response: 

The phrase has been amended in lines 245-247: " NVivo 12 software (QSR International) was used for data coding, encompassing open, axial, and selective coding. Theme development involved grouping codes into primary themes and breaking them into sub-themes.”

Reviewer’s comment

Line 276- you discuss anonymity, yet the participants' organization and their job roles are listed. This is very identifying. In line 282, its say removing job titles from data. Yet the job title is in Tables 1 and 4. Please address this important ethical issue.

Response:

We agree with the reviewer’s comment. We have removed participants' job titles from Tables 1 and 4 to address the ethical issue. The Ethical consideration section was also amended to read, “Ethics approval was obtained from the Medical Research Council of Zimbabwe, approval number (MRCZ/A/2900). All the participants gave written informed consent to take part in the interviews. Names used in this study are pseudonyms.” (Lines 260-262)

Reviewer’s comment

Results:

Table 2- shorten the questions to be key phrases/points. 

Response: 

The questions have been changed to phrases; please refer to Table 2 (lines 301-3023). 

Reviewer’s comment

Discussion:

line 568- i believe this is first time RLS is used. write it out in full in the section 'exploring opportunities for improvement' there are still results presented in the form of quotes. This discussion needs to be more tidy by summarising and discussing results rather than presenting them.

Response: 

RLS was replaced with LMIC, first mentioned in line 37 in full. Please refer to line 558. Quotations were summarised and removed from the discussion. 

Reviewer’s comment

I recommend that authors have someone external to the project review this paper in detail. There is still some room for improvement in the writing style to be more concise and scientific.

Response: 

A reviewer who published a number of papers reviewed the paper to improve its style and readability, including its scientific approach. His profile is accessible at https://www.lshtm.ac.uk/aboutus/people/timire.collins.

Reviewer’s comment: 

Reviewer #4: Thank you for the opportunity to review this interesting article which looks at medical device regulations in Zimbabwe and Tanzania. The article includes recommendations for further maturity of regulations and regulatory structures in these countries, and potentially these are applicable more widely in similar settings.

Firstly, I would like to check with the authors that they have explicit consent for the details on job title used here and especially in table 4 this is quite an identifying characteristic (just short of naming participants in many cases).

I have a number of other, minor comments, with the hope to improve readability and therefore bring forward the main learning of this manuscript.

Introduction

Line 82: “It also includes products…” what is “it” referring to?

Response:

“It” refers to medical devices, carrying over from the previous statement. The statement has been reviewed to read as, “Medical devices also include products that affect the structure or any physiological process within the human body, provided they do not achieve their principal intended action by pharmacological, immunological, or metabolic means.” (see lines 80-83). 

Reviewer’s comment

Line 104: please introduce the IMDRF acronym here and remove full name from line 109.

Response: The statement has been reviewed accordingly. The statement reads as follows, “Medical devices and IVD guidance for regulations set at the international level by the International Medical Devices Regulators Forum (IMDRF) were designed to ensure that the manufacturer who places the product on the market is responsible for all activities, safety and performance by mandating manufacturers to follow risk control based on the device's risk category (8). While regulations must not hinder economic operators, devices must meet the Essential Principles defined by the regulatory authority and detailed in the Global Harmonization Task and IMDRF guidance documents” (lines 101-107). 

Reviewer’s comment

Line 114: I have not seen the African region referred to as the “Afro” region – is this common? Perhaps use “African region” instead.

Response: “Afro” is normally used by the WHO. However, the line has been amended to read as African region: "In Africa, 40% of countries had no medical devices regulations, 32% had some, and 28% lacked data.” (lines 111-112). 

Reviewer’s comment

Line 119-120: Reference for this?

Response:

The reference has been added as shown in line 119 as follows, “Zimbabwe, a low-income country in Southern Africa, has inconsistent medical device and IVD regulations (11).”

Reviewer’s comment: 

Methods

General – authors state that COREQ is used for reporting however several things are left out: what are the interviewers' experience and training? what is the relationship, if any, between the interviewer and the participants? Any prior knowledge?

Response: 

The section has been reviewed to add more details. The section reads, “The researcher (CC) identified himself to participants during interviews, disclosing his identity, qualifications, occupations, gender, background, and training. The researcher did not have a relationship with the participants. As a Regulatory Affairs Professional for Medical Devices and IVDs with five years of experience in the field, the interviewer conducted remote interviews using Zoom or Teams. The researcher explained the purpose of the study to the participants before obtaining informed consent to take part in the study and to use an audio recorder. All the in-depth interviews were conducted in English, audio recorded, and transcribed verbatim.” (Lines 195-202). 

The other aspects relating to the study context, participant recruitment, participant characteristics, interview guide, data collection procedures, ethical consideration, data analysis, and researcher reflexivity findings, including quotations, are already included in the study. 

Reviewer’s comment

Line 143-145: The sentence that begins with “However, our study…” belongs in limitations section. Also I don’t fully understand this, please rephrase for clarity.

Response: 

The statement has been rephrased and moved to the limitations section: "The tools used in the studies were not piloted due to the small sample size, which prevented questionnaire testing and methodology piloting. Additionally, our study used a questionnaire and interviews to bypass the pilot phase and ensure implementation confidence. Our study is grounded in well-established regulatory practices, minimizing the risk of not being tested.”(lines 695-699)

The statement was added to explain why the data collection tools were not piloted before the study. 

Reviewer’s comment

Medical devices and IVD regulatory system

Line 392 – 394 – regarding the sentence that begins “However, it is reactive…” I see that the further text below is an example of what you mean by this, so perhaps you don't need this sentence? If you do decide to keep it, please rephrase it to add clarity.

Response: 

The statement was deleted and amended. It reads, “It was found that the medical devices and IVDs regulatory system in Zimbabwe was not proactive, effective, or proportional to the risk class of medical devices and IVDs.” (line 376). 

Reviewer’s comment

International standards and tools

Line 442 – On first reading, I was wondering what the participant mean that the model was updated? Now I see there is a section currently in the discussion (Line 630 – 659 and Table 5), which if it is indeed part of what you found in the study, is results and should be moved up here specifically as it helps to explain this paragraph and statement by the participant.

Response. The participant indicated that there were discussions to update the framework to include indicators of the GBT into the framework. The quotation from the South African participant was removed. The statement has been amended to read, “There are plans to revise the WHO GMRFMD. The WHO representative said, “The GBT indicators prompted discussions to update the model to include GBT + medical indicators”(WHO KI13).” (lines 427-428). 

Line 630 in the previous version of the manuscript has been amended to remove the quotes and discuss the significance of streamlining the regulatory framework as follows, “In the manuscript's discussion, it is imperative to acknowledge the crucial role of stakeholder involvement in achieving regulatory harmonization. The participation of policymakers, regulators, and economic operators is indispensable for garnering regulatory acceptance. Harmonization and convergence of medical device regulations, as seen in South Africa and Tanzania, are vital. Convergence adopts international guidelines, while harmonization standardizes technical guidelines, easing the regulatory load on manufacturers and regulators (28)” (lines 619-626). 

We think it is inappropriate to include Table 6 in the results section. The table is a compilation of recommendations from the findings in evaluating Zimbabwe's current medical devices and IVD regulations and best practices learnt from SAHPRA and TMDA. Therefore, we have decided to maintain it in the discussion section of the manuscript. 

Reviewer’s comment

The transition period to implement new regulations

Line 489: what does “rely on the status” mean?

Response: 

The quotation was incomplete. It has been corrected to refer to the previous regulatory approval by the PHLB, which was not stringent compared to the one implemented when new regulations were enacted. “ “Initially, the population had to rely on the previous regulatory approval by the PHLB before full registration was allowed. Although grandfathering was acknowledged, regulations covered all aspects, including site performance” (lines 480-482). 

Reviewer’s comment

Line 491-2: Please clarify whose statement this is - the authors of this manuscript or the participants. If this is the authors’ statement, it belongs in the discussion.

Response: The statement belongs to the author. It was deleted from the results section. Please refer to lines 478-479. 

Reviewer’s comment

Discussion

Line 630 – 659 and Table 5 – I think this belongs in the results section.

Response:

The section was edited and reads, “In the manuscript's discussion, it is imperative to acknowledge the crucial role of stakeholder involvement in achieving regulatory harmonization. The participation of policymakers, regulators, and economic operators is indispensable for garnering regulatory acceptance. Harmonization and convergence of medical device regulations, as seen in South Africa and Tanzania, are vital. Convergence adopts international guidelines, while harmonization standardizes technical guidelines, easing the regulatory load on manufacturers and regulators (28)” (lines 619-626). 

Reviewer’s comment

Line 638 – “Table 6” is below labelled table 5.

Response: 

The table was amended to Table 6. We think it is inappropriate to include Table 6 in the results section. The table is a compilation of recommendations from the findings in evaluating Zimbabwe's current medical devices and IVD regulations and best practices learnt from SAHPRA and TMDA. Therefore, we have decided to maintain it in the discussion section of the manuscript.

---

## [Decision Letter · Decision Letter 4]

18 Apr 2024

Navigating Regulatory Landscape: A Qualitative Exploration of Medical Devices and In Vitro Diagnostic Medical Devices Oversight in Zimbabwe through Key Stakeholder Perspectives

PONE-D-23-17224R4

Dear Dr. Chiku,

We’re pleased to inform you that your manuscript has been judged scientifically suitable for publication and will be formally accepted for publication once it meets all outstanding technical requirements.

Kind regards,

Adetayo Olorunlana, Ph.D.

Academic Editor

PLOS ONE

Additional Editor Comments (optional):

Reviewers' comments:

Reviewer's Responses to Questions

**Comments to the Author**

1. If the authors have adequately addressed your comments raised in a previous round of review and you feel that this manuscript is now acceptable for publication, you may indicate that here to bypass the “Comments to the Author” section, enter your conflict of interest statement in the “Confidential to Editor” section, and submit your "Accept" recommendation.

Reviewer #4: All comments have been addressed

2. Is the manuscript technically sound, and do the data support the conclusions?

Reviewer #4: (No Response)

3. Has the statistical analysis been performed appropriately and rigorously? 

Reviewer #4: (No Response)

4. Have the authors made all data underlying the findings in their manuscript fully available?

Reviewer #4: (No Response)

5. Is the manuscript presented in an intelligible fashion and written in standard English?

Reviewer #4: (No Response)

6. Review Comments to the Author

Reviewer #4: (No Response)

7. PLOS authors have the option to publish the peer review history of their article (what does this mean?). If published, this will include your full peer review and any attached files.

Reviewer #4: No

---

## [Editor Report · Acceptance letter]

29 Apr 2024

PONE-D-23-17224R4 

PLOS ONE

Dear Dr. Chiku, 

I'm pleased to inform you that your manuscript has been deemed suitable for publication in PLOS ONE. Congratulations! Your manuscript is now being handed over to our production team.

Kind regards, 

on behalf of

Associate Professor Adetayo Olorunlana 

Academic Editor

PLOS ONE